

# Implications for the Resilience of Modern Coastal Systems Derived from Mesoscale Barrier Dynamics at Fire Island, New York

Daniel J. Ciarletta[1], Jennifer L. Miselis[1], Julie C. Bernier[1], Arnell S. Forde[1]

[1]U.S. Geological Survey, St. Petersburg Coastal and Marine Science Center, 600 4th St. S, St. Petersburg, Florida 33701, USA

*Correspondence to*: Daniel J. Ciarletta (dciarletta@usgs.gov)

**Abstract.** Understanding coastal barrier response to future changes in sea-level rise rate, sediment availability, and storm intensity/frequency is essential for coastal planning, including socioeconomic and ecological management. Identifying drivers of past changes in barrier morphology, as well as barrier sensitivity to these forces, is necessary to accomplish this. Using remote-sensing, field, and laboratory analyses, we reconstruct the mesoscale (decades-centuries) evolution of central Fire Island, a portion of a 50-kilometer barrier island fronting Long Island, New York, USA. We find that the configuration of the modern beach and foredune at Fire Island is radically different from the system's relict morphostratigraphy. Central Fire Island is comprised of at least three formerly inlet-divided rotational barriers with distinct subaerial beach and dune-ridge systems that were active prior to the mid-19th century. Varying morphologic states reflected in the relict barriers (e.g., progradational, transgressive) contrast with the modern barrier, which is dominated by a tall and nearly continuous foredune and is relatively static except for erosion and drowning of fringing marsh. We suggest this state shift indicates a transition from a regime dominated by inlet-mediated gradients in alongshore sediment availability to one where human impacts exerted greater influence on island evolution from the late 19th century onward. The retention of some 'geomorphic capital' in Fire Island's relict subaerial features combined with its static nature renders the barrier increasingly susceptible to narrowing and passive submergence. This may lead to an abrupt geomorphic state shift in the future, a veiled vulnerability that may also exist in other stabilized barriers.

## 1 Introduction

Barrier coasts, including barrier islands, spits, and strandplains, front portions of every continent on Earth. Among these landforms, sandy barrier islands are commonly located along the subtropical to subpolar coasts of passive continental margins (Davis, 1994), including the east coast of North America (Leatherman, 1979a). The eastern seaboard of Canada, the United States, and Mexico contains nearly 4,300 km of barrier islands (Stutz and Pilkey, 2001), and the almost continuous stretch of barriers within the United States is among the largest reaches of barrier islands in the world (Zhang and Leatherman, 2011). Despite their ubiquity, efforts to assess barrier morphologic resilience and future evolution in the face of rising seas and increasing storm frequency/intensity are complicated by (1) their diverse present-day geomorphology, and (2) a lack of insight



regarding the relative importance of the various mesoscale (decades to centuries) drivers of morphologic change under different environmental conditions (Cooper et al., 2020; Vousdoukas et al., 2020).

Field and modeling studies have demonstrated that mesoscale barrier dynamics, defined here as barrier-lagoon behavior at decadal/centennial timescales and meter to kilometer spatial scales (Cooper et al., 2018; Sherman, 1995), are
significantly controlled by sediment accommodation and availability (Brenner et al., 2015; Ciarletta et al., 2021; Cooper et al., 2018; Psuty, 2008; Raff et al., 2018; Shawler et al., 2021a). These drivers are a function of antecedent topography (e.g., pre-transgressive surface morphology; Shawler et al., 2021a), inlet dynamics (Nienhuis and Lorenzo-Trueba, 2019), climate and vegetation (Jackson et al., 2019; Mendes and Giannini, 2015), and inherited morphology (relict barrier morphological features; Timmons et al., 2010). The latter overlaps with the concept of geomorphic capital, which is defined as sediment reserves that
must be exhausted before frontal erosion of a barrier transitions to wholesale migration (Mariotti and Hein, 2022). Mesoscale barrier dynamics also include human interventions, which have impacted coastal barriers directly and indirectly for decades to hundreds of years through manipulation of sediment input and partitioning across the entire shoreface-barrier-marsh-lagoon system (Abam, 1999; Elko et al., 2021; Hein et al., 2019; Rogers et al., 2015; Tenebruso et al., 2022; Williams et al., 2013), as well as through stabilization and destruction of barrier geomorphic boundaries, such as the backbarrier-marsh interface
(Stutz and Pilkey, 2005; Tenebruso et al., 2022).

Modeling allows for the investigation of future morphologic change based on the interaction of natural and human drivers, though such efforts are limited due to a lack of historical data to constrain input parameters. Particularly for semi-natural and developed barriers, this means that information concerning the natural balance of forces affecting system morphology must be gleaned from the geomorphic record. One approach is to decode the record of barrier state change from
relict subaerial morphology (Ciarletta et al., 2019c, 2021), a process which is more regularly applied in strandplain systems (Bristow and Pucillo, 2006; Nooren et al., 2017; Oliver et al., 2019). Barrier islands can also retain an abundance of relict features, and recent attention has been placed on their importance in illuminating past evolution and drivers of morphologic change (Billy et al., 2013, 2014; Raff et al., 2018; Shawler et al., 2019; Shawler et al., 2021b). Such efforts typically include conventional morphostratigraphic investigations in the form of core analyses and ground-penetrating radar scans, which can
provide additional sedimentological and structural information to help reconstruct past barrier environments.

Here, we use geomorphic mapping of active and remnant dune features at Fire Island, New York, USA to gain insight into both natural and anthropogenic drivers of barrier landscape change. Although Fire Island has been the subject of numerous field investigations and modeling studies (Leatherman, 1985; Lentz and Hapke, 2011; Locker et al., 2017; Schmelz and Psuty, 2022; Schwab et al., 2000; Schwab et al., 2014; Ziegler et al., 2022), little is known about the island's internal structure or the
timing of its development, especially in its central ~24 km. New geomorphic mapping and historical documentation of the island's landscape identifies locations where significant records of morphologic change are preserved. Ground-penetrating radar investigations coupled with coring and radiocarbon dating provide chronological control and paleoenvironmental information. In total, we reconstruct the evolution of Fire Island to understand differences between present and past



morphologic states, including how such differences could affect the system's future resilience and the implications for

mesoscale behavior of barrier systems globally.

## 2 Background

### 2.1 Study Setting

Fire Island is a west/southwest-oriented, 50-km long barrier island located on the south coast of Long Island, New York, USA (Figure 1). It is bound to the west by Fire Island Inlet and to the east by Moriches Inlet. Fire Island was an unbroken barrier

for 74 years (Leatherman and Allen, 1985), until Hurricane Sandy breached it in 2012 and created Wilderness Inlet 13 km west of Moriches Inlet. West of Wilderness Inlet, Fire Island is separated from Long Island by the kilometers-wide Great South Bay, which is predominantly an open water lagoon with limited fringing marsh, especially along the barrier margin. To the east of Wilderness Inlet, approximately 7 km of the island fronts the mainland Mastic Peninsula and is backed by the constricted lagoon of Narrow Bay. Further east, the backbarrier lagoon widens again, and the island fronts about 4 km of Moriches Bay,

divided from the updrift Westhampton barrier by Moriches Inlet.

Fire Island is part of a regional system of occasionally mainland-attached barriers referred to as the South Shore Beaches or Great South Beach (hist.) that extends westward from glacial outwash headlands along the southeast coast of Long Island (Leatherman, 1985; McCormick et al., 1984). The direction of elongation along the South Shore Beaches reflects an east to west net littoral transport direction, which is primarily driven by cyclonic storms tracking northeasterly through and

offshore of the Mid-Atlantic Bight (Hapke et al., 2010; Leatherman, 1985; van Ormondt et al., 2020). The area is microtidal, with a range of about 1.3 m (Leatherman, 1985).

Study area framework geology reflects Long Island's glacial origins. Nearly all Long Island surficial geology is composed of Laurentide Ice Sheet (LIS) sediments deposited during the Wisconsin glaciation (Fuller, 1914). Deposits in the study area exist within a broad glacial outwash plain that extends southward from a succession of roughly east-west oriented

moraines that define the central and northern portions of Long Island. Most notable are the Ronkonkoma and Harbor Hill moraines (Fuller, 1914), the former being active approximately 24 kya as the LIS reached its maximum extent (Figure 1; Balco and Schaefer, 2006). Digital elevation models depict two broad glacial outwash channels that descend from the Ronkonkoma moraine towards the coast in the vicinity of Fire Island (Figure 1), also identified by Fuller (1914). A channel underlying the Connetquot River appears to extend beneath the westernmost portion of the island, whereas another channel underlies the

Carmans River and aligns with Wilderness Inlet (Figure 1) and was detected beneath the barrier shoreface there (Locker et al., 2017). Portions of Fire Island that do not overlie the glacial channels are thought to rest on antecedent topographic highs which could have acted as pinning points during late Holocene transgression (Locker et al., 2003; Shawler et al., 2021a). This is supported by well logs that identified glaciofluvial sediments within just a few meters of the surface of eastern Fire Island (Schubert, 2010) and by seismic studies in the central part of the island that identified a probable submerged topographic

high—here referred to as the Central Submerged Headland (Figure 1)—potentially partly outcropping in the shoreface





(Schwab et al., 2014). The latter may act as a source of sediment to adjacent shoreface-attached ridges (Figure 1) and ultimately the subaerial barrier (Schwab et al. 2014).

Although Schwab et al. identified a potential local source of sediment to Fire Island, it is thought that most of the sand moving through the modern barrier system comes from the littoral sediment supply. From Montauk to Shinnecock, the southeasterly section of the Ronkonkoma moraine is exposed to coastal bluff erosion where it actively sources sediment to the South Shore Beaches (Leatherman and Allen, 1985). Earlier in the Holocene, ancestral barriers were further offshore, and it is thought that they derived sediment from deposits in now-submerged portions of the glacial outwash plain. The remnants of these ancient barriers are found 8 km offshore of modern Fire Island in the form of meters-thick sandy deposits arranged parallel to the coast (Sanders and Kumar, 1975). These deposits are composed of presumed lower shoreface sand that was left stranded on the continental shelf as sea level increased rapidly (Sanders and Kumar, 1975; Rampino and Sanders, 1980), possibly in association with a glacial meltwater pulse immediately preceding the 8.2-kyr global cooling event (Hijma and Cohen, 2010). Whether this ancient system survived drowning and transgressed to the position of the modern barrier remains unknown (Rampino and Sanders, 1981, 1982, 1983). Sediment cover overlying the transgressive unconformity between the modern and ancient system is thin, except for the succession of km-scale shoreface-attached ridges along the western end Fire Island (Schwab et al., 2013, 2014).

**2.2 Recent Geomorphic Change at Fire Island**

Fire Island's surface geomorphology is strongly influenced by the local alongshore transport gradient (Figure 2) and was previously divided into four distinct zones based on surface geomorphic features (Ciarletta et al., 2021). Here, we supplement previous geomorphic feature interpretations with additional insights regarding barrier behavior over the last ~200 years (Leatherman and Allen, 1985). On the updrift (eastern) end of the island, the morphology is low-relief and transgressive (Leatherman and Allen, 1985), featuring a single overwashed dune line (Figure 2a; Zone I). Conversely, the downdrift (western) end is historically elongational and contains the remnants of numerous poorly developed beach-ridge and foredune arcs (Figure 2e; Zone IV) that formed in succession with a westerly-migrating spit end (Leatherman and Allen, 1985). In the central region of the island (Figure 2c/d: Zones II and III), as many as 1 to 4 shore-sub/parallel relict foredune ridges are seen in combination with the active foredune (Lentz and Hapke, 2011). The relict dunes are generally around 2-5 meters in elevation (NAVD88), potentially indicative of an environment that was formerly subject to combinations of spatially variable progradation and amalgamation. Conversely, the modern foredune system comprises a mostly continuous ridge up to 8 m in elevation that appears to be either largely stable or aggradational, especially when considering the primarily stable to slightly progradational decadal shoreline change trends observed in the latter part of the 20th century (Allen et al., 2002).

We interpret modern foredune morphology and elevation as indicative of a relatively immobile barrier island, which is supported by the lack of inlet breaches in the central part of Fire Island since the early 1800s (Leatherman and Allen, 1985). Historically, the central region has been overwash-limited, subjecting the backbarrier to bayside shoreline erosion at a rate of 0.3 to 1.0 m/yr and resulting in island narrowing (Leatherman and Allen, 1985; Nordstrom and Jackson, 2005). Radiocarbon



dating from an interdune bog a few kilometers east of Point O' Woods demonstrates that the west-central portion of the barrier
has been relatively stable for as long as 400 years (Sirkin, 1972), and additional radiocarbon dates from relict flood
shoals/washover deposits beneath the central part of the barrier show that this section may have been near its modern position
as early as 1100 years ago (Leatherman, 1985). Moreover, Clark (1986) suggests that Fire Island as a whole may have been
relatively stable prior to the 18th century. Using age-controlled pollen data from cores taken from east of Watch Hill to
Shinnecock Inlet, Clark (1986) demonstrates the presence of mature maritime forests on Fire Island and the updrift
Westhampton barrier prior to this time, which is equated with a lack of inlet disturbance.

Conversely, historical records since the 18th century show the updrift end of the island fronting the Mastic Peninsula
has been overwashed and breached in numerous locations, and the most downdrift 8 km of Fire Island has elongated westward
since at least 1825, with spit-end shoreline is well documented in nautical charts, land surveys, and other historical accounts
(Leatherman and Allen, 1985; Ruhfel, 1971; Taney, 1961). Low relief and poorly developed recurved dunes were noted to
exist 8 km updrift of the 1825 spit-end shoreline (Leatherman and Allen, 1985; McCormick, 1984), suggesting westward
elongation of the island likely occurred prior to the 19th century. This 16-km section of elongation is believed to have
originated from the area of Point O' Woods, where the morphology transitions to a shore-subparallel succession of relict dune
ridges interspersed with mature maritime forest—the latter indicative of a long period of relatively stable conditions
(Leatherman and Allen 1985; Ruhfel, 1971; Sirkin, 1972,). This interpretation is consistent with the evolution of Fire Island
Inlet, which is thought to have migrated westward from the vicinity of Point O' Woods around the 1680s (Ruhfel, 1971;
Suydam, 1942), although no scientific investigations have previously been undertaken to validate this.

Although evidence indicates long-term barrier stability in central Fire Island, particularly in Zone III, other authors
suspected this section of the barrier was more dynamic than observed historically (Leatherman and Allen, 1985; McCormick
et al., 1984). Regardless of the exact nature and temporal framework of its stability, the relative longevity of this section of the
barrier (Figure 2c/d; Zones II and III) suggests it contains a long-term, prehistoric geomorphological record of past changes in
sediment fluxes and environmental forcing. In this study, we leveraged this stability and collected comprehensive geologic
data from zones II and III. Along with similar data from adjacent areas of Zones I and IV, we (1) determine the timing of relict
ridge and beach formation in Zones II and III, (2) identify the drivers of changes in sediment availability that influenced island
evolution, and (3) infer possible evolutionary pathways for Fire Island and other barriers in the future.

## 3 Observations and Methods

### 3.1 Geomorphic Interpretation

We first assessed central Fire Island's geomorphology using lidar digital elevation models (DEMs) from 2020 (U.S. Army
Corps, 2021) and 2014 (Brenner et al., 2016), with additional information about long-term geomorphic change derived from
historical shorelines (see Allen et al., 2002, Himmelstoss et al., 2010, and Terrano et al., 2020). This work guided subsequent
field investigations and age control analyses and provided detailed geomorphic context for previous observations. Both lidar



datasets have a horizontal resolution of 1 m and centimeter-scale vertical accuracy (Brenner et al., 2016; U.S. Army Corps, 2021).

DEMs spanning the study area were mosaicked in ArcGIS and shaded from 0 to 4 m elevation (NAVD88) to highlight the structure of relict dune ridge features, which are generally lower than the modern foredune system (see Figures 2 and 3).

Trends of relict and active dune ridges in central Fire Island were hand-digitized as line segments, highlighting the presence of former inlets as well as the general structure of relict island platforms and modern accretional environments. In combination with ground-penetrating radar (GPR) data (see next subsection), the extent of relict island platforms, former inlet fills, and the modern foredune-beach system were also hand-digitized. These features were combined with geochronological data from radiocarbon analyses to produce a morphochronological map of central Fire Island (Figure 4).

**3.2 Morphostratigraphic Investigations**

GPR data from 2021 (Forde et al., 2023) and 2016 (Forde et al., 2018a/b) were used to characterize the morphostratigraphy of specific sites. Subsurface profiles were acquired using a GSSI SIR-3000 GPR system with a 200 MHz antenna and differential GPS position control. Radar wave velocity corrections were applied to profiles based on hyperbola analyses to determine dielectric constants, rendering depth-adjusted profiles for subsequent elevation correction and interpretation. Where position

fixes were absent, elevation corrections were applied to depth-adjusted profiles by using topography from 1-m lidar DEM grids. For raw/processed data and detailed methods for 2016 profiles see Forde et al. (2018a/b); for the 2021 profiles see Forde et al. (2023).

To verify stratigraphy, characterize the subsurface depositional environments, and acquire dateable material, two sediment coring techniques were employed. Cores C1 to C4 (Figure 3b/d) were obtained using a vibracore system consisting

of a Dreyer 2-1/8" vibrator head powered by an 8-horsepower motor. Core tubes used by this system consisted of 3" diameter aluminum irrigation pipe. Cores C5 to C9 (Figure 3a/c) were obtained with an AMS 1-1/4" x 36" stainless steel sand probe, or "sand auger," which uses 1" diameter clear plastic core tubes for sediment recovery. Position control was accomplished using a differential GPS receiver. All cores were split, sampled, and described in the Sediment Core Laboratory at the U.S. Geological Survey, St. Petersburg Coastal Marine and Science Center, Florida. Core descriptions utilized Munsell soil charts

to characterize color and were photographed using a Nikon D80 digital single-lens reflex camera.

Core sections were sampled for grain-size analysis and age control based on core descriptions and comparison with GPR transects. For grain size, sediment samples were run through both a mechanical sediment sieve and a laser particle sizer, producing parallel analyses. For the sieve analysis, samples were sorted from clay to coarse sand, with no further sieving above 2 mm diameter. For the laser particle sizer, the fraction of sediment with sub 1 mm diameter was analyzed. Data shown in this

study depict the sub 1 mm grain size distributions from the laser particle sizer normalized to the total mass of each sediment sample. Raw and processed grain-size data, as well as core descriptions, images, and technical methods are available in Bernier et al. (2023).

.



### 3.3 Age Control

Age control was obtained for selected organic-rich sediment samples by accelerated mass spectrometry (AMS) radiocarbon dating—a full list of samples and calibrated ages can be found in Section 4.3. Radiocarbon samples were processed using both the organic sediment fraction and plant remains, with analyses performed by Beta Analytic, Inc., in Miami, Florida. For each sample, we report not only the conventional $^{14}$C radiocarbon age, but also the isotope ratio mass spectrometer (IRMS) $\delta^{13}$C with respect to VPDB (Vienna Pee Dee Belemnite), as well as calibrated age ranges. Calibrated ages are based on terrestrial

calibration curves from INTCAL20 (Reimer et al., 2020) using the High Probability Density (HPD) Range Method (Ramsey, 2009).

For plant remains and organic sediment fraction, we perform an environmental interpretation based on the observed/modeled relationship between $\delta^{13}$C and salinity in marsh sediments. In estuarine systems along the northeast coast of North America, decreasing salinity generally results in a progressive increase in $\delta^{13}$C depletion (Chmura and Aharon, 1995).

This occurs due to the increasing presence of plants utilizing C-3 photosynthesis rather than C-4 photosynthesis in upland environments. C-3 plants generally have $\delta^{13}$C in the range of -23 to -34 ‰, while C-4 plants range from -9 to -17 ‰ (Chmura and Aharon, 1995 c.f. Smith and Epstein, 1971). For organic sediments, this interpretation relies on the assumption of minimal contribution from algae, which display a wider range of $\delta^{13}$C values than plants across all salinity levels (Malamud-Roam and Ingram, 2001; Tanner et al., 2007). However, since we report $\delta^{13}$C for both the organic sediment fraction and plant remains,

this allows for a more robust interpretation than would be possible with organic sediment alone.

## 4 Results

### 4.1 Morphochronological Mapping

Relict dune ridge structure in central Fire Island reveals an updrift-downdrift morphologic dichotomy (Figure 4; compare b with c/d). From the Wilderness Inlet to Davis Park (Zone II; Figure 4b), the major barrier morphology consists mostly of two

large (5-8 m elevation) shore-parallel dune ridges that gradually amalgamate westward into a single dune ridge. At the western end of Zone II, around Watch Hill, there are several prominent recurved ridges preserved in the barrier interior, with evidence of seaward truncation. These recurves abruptly end at a km-wide low point in island topography partly backed by a discontinuous ridgeline (Figure 4c). Downdrift of this low point, moderate-elevation relict dune ridge successions dominate the barrier platform within Zone III.

Focusing on Zone II, historical shorelines show that the bifurcation of the foredune ridge in the updrift direction is related to inlet processes, with the eastern end of the Zone II barrier rotating seaward ('rotational barrier'—see Leatherman et al., 1982; McBride et al., 1995) after a previous iteration of the Wilderness Inlet known as Old Inlet closed in 1825, along with the closure of an immediately adjacent updrift inlet channel known as Smith Inlet in 1834 (Leatherman and Allen, 1985; McCormick et al., 1984; Figures 4b and 5). Shoreline surveys (Himelstoss et al., 2010) and historical change analyses (Allen





et al., 2002) confirm seaward progradation and amalgamation in Zone II persisted until about the 1930s, with subsequent gradual retreat and relative stability consistent with local aggradation (Allen et al., 2002; McCormick et al., 1984). This aggradational phase was interrupted by the (re)opening of Wilderness Inlet in 2012, which created a downdrift erosion shadow that has so far resulted in the destruction of the most seaward foredune ridge for about 2.5-km in the alongshore (Figure 5).

On the western end of Zone II, past inlet activity has been documented in the region surrounding Watch Hill.
McCormick et al. (1984) identified a former inlet on the updrift side of Watch Hill, herein referred to as "Long Cove Inlet" (Figure 4), which was open between 1770 and 1827 (McCormick et al., 1984), overlapping in time with Old Inlet. This created an island between the two inlets for more than half a century, which we refer to as the "Wilderness Barrier" (Figures 4b and 15). McCormick et al. (1984) also identifies another inlet on the downdrift side of Watch Hill, corresponding with a low spot west of the recurved ridges, but they could not identify any historical sources to confirm when this inlet was active. More
recently, a map from 1670 was identified as depicting an inlet at this location, adjacent to what was formerly a whaling station (Strong, 2018), and so we refer to this inlet as "Whalers Inlet" (Figure 4). Based on lidar DEMs, aerial images, and ground observations, we interpret that the Watch Hill barrier originated as a recurved spit complex that elongated from the vicinity of Long Cove and migrated downdrift in association with Whalers Inlet (Figure 6). On the updrift side of what was likely the final position of Whalers Inlet, an arcuate succession of sub-meter elevation swash ridges is present (Figure 6d). These are
distinct from larger 2-5 m elevation recurved dune ridges, and they ring a central high point, consistent with a washaround origin.

The inlet and spit complex at Watch Hill would have been part of an older iteration of the Wilderness Barrier pre-Long Cove Inlet. Its existence created a downdrift erosion shadow coincident with what we identify as an overwash-impacted transgressive dune backing most of Davis Park (Figure 6b). This reworked and discontinuous dune line is equivalent to the
heavily overwashed, low-relief ridge that presently exists on the downdrift side of modern Wilderness Inlet. We note that the former transgressive dune at Davis Park abuts a succession of relict foredune ridges (Figure 6b), with this transition defining the boundary between Zones II and III.

In Zone III (Figure 4c), the island downdrift of the Whalers Inlet erosion shadow features a succession of at least two shore-parallel relict foredune ridges fronted by a modern foredune which is amalgamated in some places with the first of the
relict ridgelines (see section 4.2 for subsurface interpretation). Overlap of the modern foredune with the relict ridge field and overall barrier narrowing increases in the downdrift direction, until relict ridges almost entirely disappear about 3 km downdrift of Whalers Inlet. Beyond this point, the barrier widens slightly downdrift, and the modern foredune is backed by relict ridges which gently recurve to the northwest for about 1.5 km before terminating at another low spot near Fire Island Pines. We interpret this downdrift section of recurved ridges as the location of prehistoric island elongation associated with the migration
of an inlet which we call "Pines Inlet." As with Whalers Inlet, the final location of Pines Inlet is preserved well in the modern morphology of the barrier, with a landward-offset ridge complex downdrift of the inferred inlet throat (compare Figure 4c/d with Figure 2d). We refer to the section of island bound by Pines Inlet and Whalers Inlet as the "Barrett Beach Barrier," named after the portion of National Park Service land near its midpoint (Figure 4c). Age control is lacking for Pines Inlet, so it is not





clear if the Barrett Beach Barrier existed as a fully independent island west of Watch Hill, although the layout of its relict ridge
system is distinct from both the extant Watch Hill and Wilderness barriers. The Barrett Beach Barrier is probably much older
than either of the updrift barriers, as the modern lagoon shoreline is eroding into the base of the rearmost relict ridges. This
process has yet to become obvious in the Wilderness Barrier despite it being relatively impervious to overwash since it began
rotating seaward two centuries ago.

West of Pines Inlet, the barrier narrows to a point about 3 km downdrift at Sailors Haven (Figure 4d). Here, the
modern barrier is backed by a single, discontinuous relict dune ridge, and the modern foredune is cut by a prominent washover
channel. Island width increases downdrift of Sailors Haven, and the modern foredune is backed by an increasing number of
relict dune ridges that recurve and splay gently towards the northwest and become truncated by the lagoon shoreline. The relict
ridges abruptly terminate at Point O' Woods, beyond which the barrier topography into Zone IV is dominated by a single
foredune ridge backed by the remnants of low-relief recurved ridges associated with spit-building as Fire Island Inlet migrated
westward. We refer to the section of island between Pines Inlet and the possible origin point of Fire Island Inlet as the "Sailors
Haven Barrier." As with the Barrett Beach Barrier, there is very little fringing marsh behind the Sailors Haven Barrier, and the
lagoon shoreline is in many places cutting directly into the base of relict ridges, indicating this section of island has likely
existed in its present location for centuries.

### 4.2 Subsurface Stratigraphy

Ground-penetrating radar scans within Zones II and III, specifically in the Wilderness, Watch Hill, and Sailors Haven barriers
(see Figure 3 for layout), confirm surface geomorphological interpretations and provide additional details on the cross- and
along-shore structure of the island. Figure 7 shows the cross-shore structure of the Wilderness Barrier from the lagoon edge to
the upper beach in Zone II at Ho-Hum Beach. Core C4, indicated in yellow, was acquired from the rear dune ridge, which was
likely the primary foredune when Old/Smith Inlet closed in 1834 (see Allen et al., 2002). The steep ocean-dipping reflections
seaward of C4 confirm that progradation took place from this point to the shoreline's maximum seaward position in 1933
(Figure 5). Additionally, gently dipping reflections landward of C4 are consistent with washover deposition. These
observations imply that the updrift portion of the Wilderness Barrier was formerly transgressive and reached its most landward
position in the early 19th century.

Figure 8 shows the alongshore subsurface structure downdrift of Ho-Hum, in the Watch Hill Barrier section of Zone
II. In the updrift portion of this transect, reflections dip updrift, consistent with shore-subparallel erosion propagating from
Long Cove Inlet (Figure 8, 700+ m). Further west, reflections become horizontal before reversing direction and dipping
downdrift (Figure 8, 300 to 500 m). We associate this convex structure with a recurved barrier platform, and we later confirm
that this platform is truncated in the cross-shore direction via core analysis (see Section 4.3). Another set of convex reflections
partly onlap the tapering end of this "truncated beach" platform (Figure 8, 200 to 400 m), and we interpret these as comprising
a discrete and bulbous recurved barrier platform which we refer to as the "The Lobe." The horizontal extent of this feature is
discernable in Figure 6. The upper surface of the lobe is dominated by swash-aligned washaround ridges encircling a central





high point about 200 m landward of the alongshore transect. Near the middle of the lobe, a borehole (S125990) from a previous study (Schubert, 2010) indicated the presence of the underlying transgressive surface at -5 m elevation. Reflections dipping downdrift from the truncated beach and partly underlying the lobe become horizontal at this elevation and indicate the lobe is
directly overlying the transgressive surface. Downdrift of the lobe, sharply concave reflections penetrate to an elevation of -6 to -7 m, consistent with a former inlet throat that scoured deep enough to excavate antecedent topography (Figure 8, 100 to 200 m). This feature likely marks the final position of Whalers Inlet before closure. Corresponding surface morphology depicts a remnant inlet slough abutting a downdrift transgressive ridgeline consistent with an erosion shadow (Figure 6b).

The transition from Zone II to III is marked by the appearance of surface successions of multiple relict sub/parallel
dune lines beginning just west of Davis Park in the Barrett Beach Barrier (Figure 4c). Figure 9 depicts a cross-shore profile through this ridge field, revealing an underlying barrier platform characterized by seaward-dipping reflections signifying past beach progradation. Above these progradational packages, discrete relict dune ridges are evident. The modern foredune is shown to be partly amalgamated against the most seaward relict ridge and is double the height of older dunes, which could be indicative of relatively prolonged barrier/beach stability or enhanced subaerial sediment availability.

Similar subsurface morphology is observed in the Sailors Haven Barrier in both the central (Figure 10) and downdrift (Figure 11) parts of the platform. The central area (Figure 10) is comprised of tightly packed seaward-dipping reflections overlain by a succession of short, partly buried dunes and/or berms. These short dunes are dwarfed by the modern amalgamated foredune, which rises to a heigh of 4 m over the otherwise relatively flat barrier surface. At the downdrift end of the platform (Point O' Woods), the subsurface structure is dominated by seaward-dipping reflections, and in spots is overlain by relict dunes
that are buried up to 2 m below the modern barrier surface (Figure 11). Approximately 75-100 m landward of the modern foredune, reflections depict a shallow swale approximately 100 m in width which is filled by washover. Along the seaward margin of the swale, a 50 m-wide and 1.5-2 m thick interval of convex reflections is consistent with a buried, overwash-impacted dune. A succession of progradational packages that underlie the modern foredune offlap from this feature. In total, these results suggest that Point O' Woods sustained older and younger episodes of progradation, possibly separated by an
episode of beach transgression that at one point eroded the barrier to a position ~75 m landward of the modern foredune.

Alongshore GPR transects at Point O' Woods depict only gently downdrift-dipping reflections consistent with the recurvature of surface dune ridge traces (Figure 4d). Just downdrift in Zone IV, and beyond the mapped limit of the Sailors Haven Barrier, alongshore reflections dip relatively steeply in both the seaward and downdrift directions, consistent with past spit growth (Figure 12). This supports historical accounts and morphological impressions which indicate that the 16 km of Fire
Island within Zone IV comprise a spit that has grown westward from the vicinity of Point O' Woods.

**4.3 Lithology and Environmental Interpretation**

The lithology of Zone I can be characterized by core C5, recovered 2.5 km east of Wilderness Inlet at Smith Point (Figure 13; see Figure 3a for core location). Collected at the backbarrier-marsh interface, the bulk lithology of the core is consistent with a succession of washover deposits. It primarily features interbedded coarse to medium sand, but also at least one distinct unit



displaying normal bedding (fining upward from coarse to medium sand). Despite being just a few meters seaward of the marsh fringe and penetrating more than a meter below the land surface, no marsh units were recovered. Since multiple marsh intervals were identified in cores from adjacent barriers and are believed to coincide with quiescent periods between extreme storms (Bennington and Farmer, 2015), their absence in our core likely indicates frequent washover consistent with the transgressive morphology of Zone I.

330        In Zone II, the lithology from core C4 confirms our morphostratigraphic interpretation that the former foredune of the Wilderness Barrier was in this location in the early the 19th century (Figure 13). The upper 158 cm of the core is characterized by laminated fine-medium aeolian sand—finer than recovered in any other core. Recovered dune sand overlies a 12-cm interval of medium-coarse sand and occasional shell fragments interpreted to be former beach/backshore. This lithology correlates directly with the adjacent cross-shore GPR profile seen in Figure 7, which shows sub-horizontal reflections

consistent with aeolian deposition overlying seaward-dipping reflections diagnostic of a former beach.

        The lithology from core C9, recovered from the throat of Long Cove Inlet, is consistent with inlet closure, and dating confirms sediments overlap in time with when Love Cove Inlet is believed to have filled in, ultimately reconnecting the Wilderness and Watch Hill barriers (Figure 13). Specifically, the bottom 4 cm of the core consists of organic-rich sand, which is overlain by a deposit of interlaminated coarse and medium sands and a veneer of loose, medium sand with organic-rich

horizons. Accounting for surrounding surface morphology (Figure 6), we interpret this sequence as a vegetated inlet slough infilled by washover, the surface of which was subjected to aeolian reworking. Radiocarbon dating of the sediment component of the organic sand unit returned an age of 90 ± 30 yrs BP, whereas plant remains within this unit returned 120 ± 30 yrs BP. Including calibrated age ranges (Table 1), this overlaps with the historically documented timeframe for Long Cove Inlet's closure in 1827 CE (McCormick et al., 1984). Additionally, plant $\delta^{13}$C was measured at -16.1‰, which in the U.S. northeast

is consistent with a low marsh environment (Chmura and Aharon, 1995) and suggests sediment at the bottom of the core was at least briefly under the influence of the Great South Bay before burial by washover.

        The elongational spit complex of the Watch Hill Barrier was sampled by Cores C8, C7, and C6 (Figure 6). C8 targeted the seaward margin of the relict foredune on the "truncated beach" platform (Figure 8), while C7 and C6 targeted landward and seaward areas on "The Lobe" immediately downdrift. In general, core lithologies and age control support the interpretation

that both platforms experienced seaward truncation in association with the erosion shadow of Long Cove Inlet. Relict beach lithology is found in the upper part of core C8. Sediments between 23 and 43 cm consist of alternating bands of dark reddish brown heavy-mineral sand and clean grayish sand, with thinner alternating bands observed around 60 cm. Since a similar sequence was found on the modern beach after a late-season nor'easter impacted Fire Island on April 18 and 19, 2022, just two days before C8 was acquired (Figure 14), these sediments likely represent post-storm beach rebuilding. Deeper lithological

units within C8 demonstrate these beach sediments also reflect shoreline transgression on the "truncated beach" platform. Below the beach beds is a half-meter thick sequence of medium to coarse sand overlying two distinct layers of hemic peat separated by a thin sand bed. From the bottom up, this lithology is interpreted to record an episode of prolonged quiescence in a protected backbarrier environment—probably a drowned interdune and/or inter-spit swale—which was interrupted by





relatively rapid emplacement of washover due to shoreline retreat. Radiocarbon dating of organic sediments in the upper and

lower peat layers return ages of 280 ± 30 and 260 ± 30 yrs BP, respectively, while plant material produces ages of 130 ± 30 and 180 ± 30 yrs BP.  As with Core C9, plant $\delta^{13}$C (Table 1) is consistent with a low marsh setting, while organic sediment $\delta^{13}$C is more consistent with a high marsh setting. In both the plant and organic sediment fractions, $\delta^{13}$C becomes less depleted in the up-core direction, which suggests increasing salinity prior to washover emplacement. Subsequently, we posit that the youngest plant remains could represent the timeframe just prior to burial by washover. This assumption combined with an age

comparison of plant remains for C8 and C9 (Table 1), suggest that the relict beach sediments in C8 reflect shoreward truncation of the Watch Hill Barrier resulting from the erosion shadow of Long Cove Inlet. This is further supported by a Coast and Geodetic Survey chart that shows a remnant indentation of the shoreline downdrift of the former Long Cove Inlet in 1835 (Leatherman, 1989).

Cores C6 and C7 (Figure 13) were used to explore the nature of "The Lobe" (Figure 6) discussed in previous sections.

C6 penetrated the lobe at a more seaward position relative to C7, with its lithology comprising a thin veneer of likely wind-reworked sediment overlying a thick sequence of massive to faintly bedded medium to coarse sand interpreted as washover. This is consistent with alongshore GPR data (Figure 8) which indicate the relict lobe surface near C6 is mantled by ~2 m of transgressive sediment packages. With Core C7, we sought to recover sediment from the crest of a washaround ridge identified in aerial imagery on the lobe (Figure 6b/d). The top 23 cm of recovered section reveals a sequence of faintly laminated medium

to coarse sand, which we equate with the ridge structure. Directly below this interval is a 24-cm thick unit of coarse sand, with mm- to cm-scale mud balls immediately below the upper contact. We interpret this unit as reworked washover and marsh sediments flanking the subaerially exposed portion of the lobe. Finally, beneath the coarse sand interval is a sequence of centimeter-thick beds consisting of, from top to bottom, hemic peat, a thin bed of fine to medium sand, hemic peat, and more fine to medium sand, which we interpret as backbarrier marsh that was episodically buried by washover (Figure 13).

Radiocarbon dating of organic sediments in the upper and lower peat layers return ages of 140 ± 30 and 270 ± 30 yrs BP, respectively, while plant material produces ages of 30 ± 30 and 180 ± 30 yrs BP. Ages for the lower peat layer in C7 are comparable to the upper peat layer of C8, and they are also at the same elevation. Intriguingly, while plant-derived $\delta^{13}$C in the upper peat layer of C8 is indicative of a low marsh environment, C7's lower peat layer is relatively $\delta^{13}$C depleted and consistent with a brackish fringe setting (Table 1). Taken together, this could reflect initial narrowing of the Watch Hill barrier in the

updrift direction as a response to the opening of the Long Cove Inlet.

In Zone III, cores C1, C2, and C3 paralleled GPR line 18 (Figure 11). Core C1 penetrated the barrier's lagoon beach, C2 penetrated the interior relict dune system, and C3 targeted the area of the buried swale (Figure 13). The lithology and age of these cores illuminate a complex and relatively long sequence of morphologic change. For example, C1 records an entire cycle of barrier emplacement and drowning. Starting at the bottom, the core features an interval of medium to coarse sand that

matches with seaward dipping reflections recorded in GPR data (Figure 11), suggesting a beach origin. Above the inferred beach sand lies a unit of deformed medium to coarse sand that we equate with the base of a relict dune, or maybe a deposit akin to a beach ridge as opposed to a true aeolian dune. 39 cm of medium sand overlying this unit contains occasional root





fragments and is likely the preserved surface of the rear dune flank. This is corroborated by a landward-dipping reflection seen in the GPR data (Figure 11).

Overlying the preserved dune surface at C1 is a 31-cm thick peat interval which is overlain by a 34-cm thick washover deposit. Radiocarbon dating of the base of the peat returns an organic sediment age of 330 ± 30 yrs BP, while dating of plant remains returns an age of 140 ± 30 yrs BP. We assume the large discrepancy in age to represent contamination of older sediments with younger plant remains, but the difference is useful for inferring the longevity of the environment that produced the peat. Both the organic sediment and plant remains demonstrate $\delta^{13}$C values below -28 ‰, consistent with a freshwater

terrestrial origin in an upland bog and implying this environment persisted for multiple centuries. We note that this timeframe also matches the reported age of bog sediments analyzed by Sirkin (1972) in the nearby Sunken Forest, which yielded a date of 250 ± 80 yrs BP. It is unclear precisely when the overlying washover was deposited, but soil horizons in the top 14 cm of the washover show a terrestrial environment may have been present for some time after bog burial. Above the washover is a medium sand interval with a distinct dark brown color and organic laminae which represents the modern lagoon beach as it

actively truncates former barrier deposits.

     Similar to C1, core C2 reached the underlying surface of inferred beach/backshore sediments at an elevation of -1.5 m NAVD88. Fine to medium sand overlying these sediments are inferred to have a dune origin. Core C3 failed to penetrate as deep as C1 or C2, it but did penetrate the washover deposit seen in our GPR data (Figure 11). The deposit was found to be ~1.5 m thick and overlying 18 cm of peat lying atop a unit of medium sand. This washover may be partly sourced from the

remnants of a low, transgressive dune identified in the subsurface (Figure 11). Though C2 lacked age control, a radiocarbon sample taken from the peat unit in C3 produced the oldest ages reported in this study, with organic sediment dated to 690 ± 30 yrs BP and plant remains dated to 460 ± 30 yrs BP. As with the radiocarbon sample from C1, comparatively young plants presumably rooted in older organic sediments. Both the organic sediment and plant remains also demonstrate $\delta^{13}$C values below -25 ‰, again consistent with a long-lived upland bog; in this case, occupying a large interdune swale. The age of 690

yrs BP at C3 implies that relict beaches and ridges landward of this location (e.g., at C1 and C2) are even older. Assuming the date of 460 ± 30 yrs BP is representative of plants that grew closer to the end of the peat's deposition, this sets a maximum age on the overlying washover and subsequent transgressive episode recorded in subsurface morphology.

## 5 Discussion

### 5.1 Timeline of Barrier Change at Fire Island

Our results demonstrate that the pre-20th century landscape of Fire Island featured successions of moderate-elevation progradational foredunes and low-elevation overwashed dunes distributed across a series of inlet-separated rotational and elongational barriers through time (Figure 15). Our investigation confirms some existing hypotheses about how the island evolved over the last six centuries. GPR and age-control data around Point O' Woods suggests the western spit of Fire Island (Zone IV) began elongating from that location sometime after the late 16th century, which corresponds with the historical



timeline (Ruhfel, 1971; Suydam, 1942; Figure 15c/d). Our GPR also confirms the presence of an inlet at Davis Park, which was inferred by McCormick (1984) and more recently documented from a historical map (Strong, 2018; Figure 15d/e).

Our data also reveal new information. Clark (1986) suggested Fire Island was generally a more stable, inlet-free barrier before the 18th century based on the inferred presence of mature maritime forest communities along the eastern half of the island prior to this time. However, our analysis suggests inlet activity was instead focused on the western side of the island

between Watch Hill and Point O' Woods (Figure 15d/e), coinciding with a data gap (Clark, 1986). We demonstrate not only the presence of the alongshore-migratory Whalers Inlet around Watch Hill, but also the presence of another inlet near Fire Island Pines. The impact of the latter on island morphology may be recorded at Point O' Woods, where GPR profiles and radiocarbon dating suggests seaward truncation of the barrier occurred around or just after 1500 CE (Figure 15e).

This synthesis of present and past analyses shows that central Fire Island was subjected to spatially variable and

sustained inlet activity between the 16th and early 19th century that drove alongshore ecogeomorphic heterogeneity. This contrasts contemporary central Fire Island, which is dominated by a relatively static and continuous high-foredune system. The change in barrier morphologic state from a diverse and dynamic landscape to a relatively stationary setting was suspected by earlier researchers (Leatherman, 1989; McCormick, 1984), but age control and morphostratigraphic relations among the various landscape components of the barrier were not understood well enough to conceptually describe the island's evolution.

By confirming impressions of and providing chronological control for past barrier behavior, we can move beyond local characterization and explore how inherited morphology and the geometry of the barrier's modern beach and foredune might impact the resilience and evolution of the island in the future. Additionally, we can explore the implications of these findings for other barriers, as well as what part human interventions play in the fate of these systems based on past behavior. However, we first discuss the drivers of sediment availability in the past Fire Island system to understand why the modern system appears

as it does.

## 5.2 Inlets as Drivers of Barrier Morphologic Change: Then and Now

Changes in sediment availability are a primary driver of geomorphic complexity in barrier systems (Brenner et al., 2015; Cooper et al., 2018; Psuty, 2008), and this work shows there were significant spatiotemporal variations in the sediment budget of Fire Island over the last 700+ years. All the relict barriers identified in this study—Wilderness, Watch Hill, Barrett Beach,

and Sailors Haven—show evidence of differential progradation and transgression, as well as elongation and shortening. Most importantly, we demonstrate that these changes are related to the opening and closing of inlets. This result is consistent with observations of modern barriers which show that inlets can act as the primary driver of decadal-scale sediment redistribution, as observed in the Malpeque barriers of eastern Canada (Armon and McCann, 1979). Over longer timescales, even where overwash and other sediment transport mechanisms are active, sediment moved by inlets can add up to most of the sand volume

contained within individual barriers over the course of hundreds to thousands of years (Bartberger, 1976; Leatherman, 1989 c.f. Leatherman, 1987 and 1979b).



Our investigation shows inlet-mediated changes in the distribution of barrier sediments and destruction/creation of geomorphic capital (see Mariotti and Hein, 2022) were occurring at decadal to sub-centennial timescales and over kilometer-scale reaches. For example, radiocarbon dating of peat layers buried by washover deposits in the Watch Hill barrier confirms

that the entire beach face of this former spit complex underwent significant transgression due to the opening of Long Cove Inlet in the early 19th century. Our geomorphic and lithostratigraphic analyses, combined with historical shoreline change analyses from Allen et al. (2002), also confirm the closure of Old and Smith Inlets resulted in the seaward rotation of the entire 8-km shoreline of the Wilderness Barrier from the 1830s to the 1930s, creating the bifurcated dune system observed today.

At Point O' Woods, our results show inlet-mediated changes in barrier morphology have occurred at the island-scale
for hundreds of years. The formerly progradational landscape at Point O' Woods, likely reflecting past seaward rotation predating 690 yrs BP, was disrupted by shoreline transgression probably resulting from updrift inlet activity between 460 and 140 yrs BP. This may correspond with trends in the plan-view geometry of relict ridges that suggest there was an inlet at Fire Island Pines, midway between Davis Park and Point O' Woods (Figure 4c/d). Additionally, Fire Island's kilometer-scale elongation in Zone IV is believed to have begun around the 1680s (Ruhfel, 1971; Suydam, 1942). Since this is within the 460
to 140 yrs BP window, it implies there could be a relationship between the erosion at Point O' Woods and the initiation of downdrift spit growth. We note that inlets can also drive the liberation of sediments from antecedent topography (Shawler et al., 2019, 2021a), especially where this topography is relatively close to the surface. The spatial coincidence of previous inlets—e.g., Whalers Inlet (Figure 8)—and the Central Submerged Headland identified by Schwab et al. (2014) suggests that inlets excavated sediment that could have contributed to kilometer-scale island elongation in Zone IV.

Given that Fire Island was once extensively modified by inlets, the question is: why not today? The lack of inlet processes on modern Fire Island may relate to human development. Starting in the late 19th century, as interest in the island grew to include the establishment of communities and parks, it became progressively subject to coastal engineering. With the exceptions of Wilderness Inlet and Moriches Inlet, which was naturally formed in 1931, every breach that has developed in Fire Island since the 19th century has been mechanically closed. These include a recent breach through the barrier at Smith
County Park (Zone I) after Hurricane Sandy (Bilecki, 2020), as well as multiple breaches in the eastern end of Fire Island caused by the 1938 Long Island Hurricane (Howard, 1939). The two inlets currently bracketing the island, Fire Island Inlet and Moriches Inlet, were stabilized in the early-mid 20th century to prevent migration and closure (Leatherman and Allen, 1985; Ruhfel, 1971). Large-scale dune nourishment projects have also been undertaken, including efforts by the Works Progress Administration (WPA) to reconstruct 68 miles of dunes along the Suffolk County portion of the South Shore of Long
Island beaches in 1939 (Morang, 1999). Studies have shown that even moderately tall (3 m) constructed dunes limit natural overwash processes (Schupp et al., 2013), thus limiting inlet activity that might occur in areas of low/overwashed dune topography (McCormick et al., 1984). The net effect is that morphologic change affected by inlets, at least on the spatiotemporal scales that Fire Island once permitted, may no longer be possible.



**5.3 Inferring Future Morphological Evolution: Knowns and Unknowns**

The impacts of human interventions and climate change in coastal barrier systems are creating scenarios for which there may be no historical analogs, either ecological (Williams and Jackson, 2007) or purely morphological (Ciarletta et al., 2019a/b; Magliocca et al., 2011; Rogers et al, 2015). The combined influence of geomorphic capital and human stabilization presents a new and unknown challenge for coastal management, particularly since the volume and distribution of subaerial and subaqueous sand reservoirs present in modern barrier systems is loosely quantified at best. There is also limited understanding

about the natural rates of change that are possible in modern systems, mainly due to a lack of data about past barrier evolution for specific systems over decadal to centennial timescales. Even where such detailed assessments exist, as now does for Fire Island, future biogeomorphological resilience will depend on the interaction of modern and relict morphology, which is a dynamic that is not well understood. For example, future biogeomorphological changes modeled at Fire Island demonstrate considerable uncertainty because the modern beach and foredune reflects decades of shoreline stabilization efforts that are

likely masking natural morphologic shifts (Rice, 2015; Ziegler et al. 2022, c.f. Armstrong and Lazarus 2019). This masking effect is obvious in other barrier systems from dune scale to island scale. Beach and dune nourishment amount to artificially replenishing capital (Mariotti and Hein, 2022), and in some cases, this has resulted in narrow barriers resisting morphological state changes that wider barriers have succumbed to over just a few decades. As an example, Hog Island (Virginia, USA), a natural barrier island, was as much as 2 km wide, but temporarily reverted to a narrow, transgressive state over the course of

50 years (Robbins et al., 2022). Conversely, narrow stretches of the Bogue Banks (North Carolina, USA) that were naturally evolving towards a transgressive state have resisted morphological state change for decades due to anthropogenic dune maintenance activities and mechanical closure of breaches (Timmons et al., 2010), which contributes to uncertainty in our ability to apply both conceptual and numerical models to predict future evolution.

        In addition to interactions between geomorphic capital and human alterations, understanding of mesoscale coastal

behavior is further complicated by other barrier landscape controls that may interact with inlet activity, including the degree to which these processes may change the distribution and frequency of inlet formation and closure. In particular, spatiotemporal variations in the rates at which dunes accumulate sediment, as well as the maximum heights they achieve, could directly affect barrier vulnerability to breaching (McCormick et al., 1984). Such variations are themselves the result of complicated interactions among biophysical processes, and models indicate morphological bistability of either low or high dune states can

sometimes occur among similar combinations of storm magnitude/frequency and vegetative forcing, and potentially within the same barrier (Goldstein and Moore, 2016). Another factor influencing inlet activity is the distribution of sediment in the shoreface and inner continental shelf. Along Fire Island, there is an east-west dichotomy in the availability of inner shelf sediments which may have enhanced or even dominated the updrift-transgressive/downdrift-accretional trend in barrier surface morphology over centennial and longer timescales (Schwab et al., 2013, 2014). Yet, it is unclear whether inner shelf sediment

distribution set the conditions for where inlets formed or is itself reflective of past inlet activity.



Regardless of the processes interacting or competing with inlet activity, our results suggest that if Fire Island were capable of sustaining inlets at the decadal scale—as it once did—it would display greater alongshore variability in subaerial morphology, likely leading to more heterogeneous responses to storms and sea-level rise, as well as an increased diversity of habitats. Instead, as sea level increases, the modern island with its nearly continuous foredune system is likely to undergo a combination of gradual frontal erosion and passive drowning of the interior, with eventual transition towards a flatter and narrower state (Ziegler et al. 2022)—the starting phases of which are already evident (Nordstrom and Jackson, 2005). We hypothesize that Fire Island's resilience will continue to decline due to the relatively great height of the modern foredune, which studies have indicated could create a feedback loop of island narrowing and marsh destruction caused by a lack of washover deposition in the backbarrier (Dolan, 1972; Lorenzo-Trueba and Mariotti, 2017, Miselis and Lorenzo-Trueba, 2017; Magliocca et al., 2011; Rogers et al, 2015). Loss of morphological resilience is likely to be exacerbated by reserves of naturally available sediment in reaches of relict progradational dunes, which could enhance Fire Island's short-term resistance against landward migration but ultimately increase the island's long-term committed retreat (Mariotti and Hein, 2022). This process is evident at Ho-Hum Beach, where downdrift erosion related to Wilderness Inlet has partly destroyed the modern foredune but not resulted in any changes to either the backbarrier shoreline or the relict dune line (Figure 7). We note that the effect of shallow antecedent topography at Ho-Hum Beach (Figure 7) may also be a contributing factor in pinning the barrier's position at that location, which may induce further retreat hysteresis.

In the most extreme scenario, the combination of a tall, anthropogenically influenced foredune, an abundance of geomorphic capital, and the presence of shallow antecedent topography could lead to island instability in the future, in some parts of Fire Island and possibly elsewhere where similar conditions exist. Modeling studies show that lags in barrier response to sea-level rise tend to result in a rapid stepping back when washover begins to consistently reach the backbarrier (Ciarletta et al., 2019a; Shawler et al., 2021a). This can result in barrier drowning due to a combination of subaerial sediment loss to an over-deepened lagoon and partial abandonment of the lower shoreface (Ciarletta et al., 2019a; Lorenzo-Trueba and Ashton, 2014). Such a scenario does not also consider the possibility of an increase in storminess in combination with an increase in rate of sea-level rise, which process-based modeling has demonstrated can result in very rapid (decadal-scale) drowning of barriers due to a failure of post-storm sediment recovery to balance losses from subaerial sand reservoirs (Passeri et al., 2020). Even if this extreme scenario proves unrealistic for Fire Island, an outcome where the island simply maintains a thinner, alongshore-dominated geometry will be detrimental to mature ecological communities. This is already being realized in the island's maritime forests as they gradually drown and erode (Art, 1976; Sirkin, 1972).

It is important to note that management is not the same everywhere at Fire Island. In designated wilderness areas, such as that between Watch Hill and Smith Point, the island can breach and form inlets (e.g., Wilderness Inlet), and some return to inlet-mediated barrier rotation has been observed. However, it is unclear whether discrete areas lacking human intervention are enough to substantially alter the island's evolutionary trajectory. As of early 2023, Wilderness Inlet appears to be closing after ~10 years, a relatively short lifespan when compared with Long Cove Inlet and Old Inlet, which persisted for 50+ years and altered the morphology of the island over much larger reaches. This shortened lifespan likely resulted from



updrift nourishment activities, which can interrupt natural inlet evolution due to elevated updrift sand fluxes (e.g., Ludka et al., 2018). Though further research is needed, this behavior implies that variability in management and geomorphic capital increasingly become a secondary control on barrier geomorphic evolution as the overall ability to sustain inlets diminishes in tandem with increasingly large and overwash-resistant dunes.

Finally, the history of Fire Island's human development follows a pattern that is similar to other barriers in the region
(Tenebruso et al., 2022) and beyond (Dolan, 1972; Seminack and McBride, 2015), and there is some evidence of such systems previously experiencing a greater distribution of inlet activity than at present (e.g., Assateague Island, Maryland—Seminack and McBride, 2015; Northern Outer Banks—Mallinson et al., 2010). Because modern coastal management practices often seek to stabilize existing inlets and prevent new inlet formation, one of the most significant drivers of decadal-centennial barrier geomorphic variability is limited during a time when changes to other drivers (e.g., sea-level rise and storm
frequency/intensity) are more uncertain than ever. A dearth of inlet activity potentially promotes a decadal-scale loss of geomorphic resilience and may also alter the longer-term retreat behavior of barriers through the restriction of flood-tidal shoal deposition, which provides a platform for barrier migration and stabilization, as well as a source of sandy sediment during future transgression (Nienhuis and Lorenzo-Trueba, 2019). At a global scale, this may be promoting a scenario of future barrier destabilization and possible drowning that becomes increasingly challenging to avoid beyond centennial time horizons.

**6 Conclusions**

We found that the central region of modern Fire Island comprises a set of at least three formerly inlet-divided rotational barriers with distinct subaerial beach and dune-ridge systems that were formed by differential progradation and transgression. In particular, the central-eastern portion of the barrier reflects the most recent episode of island-scale inlet-mediated coastal change, having been a rotational barrier as late as the early 19th century. Meanwhile, the central-western section of Fire Island
preserves a long-term record of geomorphic change, revealing cycles of inlet-associated progradation and transgression stretching back 700+ years.

In contrast to its past evolution, Fire Island has seen a decrease in sustained inlet activity and is fronted by a largely stable and nearly continuous foredune. We interpret this shift in morphodynamic state as a response to human alterations and suggest that the barrier is approaching a geomorphic tipping point. Specifically, lack of landward sediment transfer and loss
of ability to generate new geomorphic capital is amplifying bay erosion and encroachment of the barrier platform, which is gradually depleting relict sand reservoirs and priming the island for a rapid state shift to transgression and possible drowning in the future. We emphasize that this process may make variations in sediment management along the island ineffective in changing evolutionary trajectories, although additional research is needed to explore this here and in other barriers.

Comparison of our findings at Fire Island with other barriers will also be needed to understand the range of rates at
which inlets naturally open and close, as well as the rates at which they alter sediment distribution across the combined shoreface-barrier-backbarrier continuum. This will help isolate the relative importance and timescales of inlet activity across

a spectrum of barriers, allowing for a more robust quantification of barrier vulnerability in the context of human development and other anthropogenic impacts. Ultimately, such endeavors could help prioritize where management activities can be altered to promote future resilience.

**7 Data availability**

Lithological descriptions, grain size analyses, core photos, core X-rays, radiocarbon data, geospatial control, and acquisition/processing methods relating to sediment samples can be found in Bernier et al. (2023). See: https://doi.org/10.5066/P91P1T88.

Ground-penetrating radar reports, scans, navigation files, and acquisition/processing methods are available via Forde et al.
(2018a/b) and Forde et al. (2023). See: https://doi.org/10.3133/ds1078, https://doi.org/10.5066/F7P84B1P, and https://doi.org/10.5066/P97YW2UL.

**8 Author contributions**

D.C. conceptualized the investigation with J.M., and executed the investigation with J.B. and J.M. Data analysis and curation was undertaken by D.C., J.B., and A.F. The original draft of the manuscript was written by D.C. and J.M., with critical feedback
and editing from J.B. and A.F. All authors agreed on the final draft.

**9 Competing interests**

The authors declare no competing interests. Any use of trade, firm, or product names is for descriptive purposes only and does not imply endorsement by the U.S. Government.

**10 Acknowledgements**

We would like to acknowledge the considerable planning and assistance received in making this work possible, particularly during a pandemic. For assistance rendered in data acquisition and survey planning, we want to thank the U.S. Geological Survey (USGS) New York Water Science Center, including Mike Noll, Bill Capurso, Tony Chu, Chris Schubert, and Ron Busciolano, among others, as well as the staff of the National Park Service at Fire Island National Seashore, including Jordan Raphael, Kelsey Taylor, Jason Demers, and Mike Bilecki. For assistance rendered in the laboratory, we want to thank our
colleagues Nancy DeWitt (core sampling, equipment training/loading), Cheyenne Everhart (grain size analysis), Jessica Jacobs (core x-rays), and Noreen Buster (core sampling). We also want to acknowledge the Point O' Woods Association and the Village of Bellport Beach for providing access to field sites. This work was made possible through the USGS Mendenhall Research Fellowship program and the USGS Coastal and Marine Hazards and Resources Program.



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

**Table 1. Radiocarbon samples obtained from core sites on Fire Island. Sample ID indicates depth with respect to land surface. Inferred environment based on isotope analysis of measured/modeled values from Chmura and Aharon (1995).**

| Sample ID | Lab No. | Material | Conventional Age | Calibrated Date / Age | | IRMS δ¹³C | Inferred Environment |
|---|---|---|---|---|---|---|---|
| C1, 121-123 cm | 621034 | plant material | 140 +/- 30 BP | (58.2%) 1797 - 1944 cal CE<br>(37.2%) 1671 - 1779 cal CE | (153 - 6 cal BP)<br>(279 - 171 cal BP) | -28.3 o/oo | Long-lived upland bog / interdune swale or high barrier flat; overlying stratigraphy indicates buried by washover |
| | **638248** | **organic sediment** | **330 +/- 30 BP** | **(95.4%) 1480 - 1640 cal CE** | **(470 - 310 cal BP)** | **-28.1 o/oo** | |
| C3, 99-101 cm | 621035 | plant material | 460 +/- 30 BP | (95.4%) 1412 - 1471 cal CE | (538 - 479 cal BP) | -25.3 o/oo | Long-lived upland bog / interdune swale; overlying stratigraphy indicates buried by washover |
| | **638249** | **organic sediment** | **690 +/- 30 BP** | **(65.5%) 1272 - 1317 cal CE**<br>**(29.9%) 1360 - 1388 cal CE** | **(678 - 633 cal BP)**<br>**(590 - 562 cal BP)** | **-26.6 o/oo** | |
| C7, 51-56 cm | 633957 | plant material | 30 +/- 30 BP | (35.8%) 1867 - 1917 cal CE<br>(31%) 1810 - 1862 cal CE<br>(28.6%) 1694 - 1725 cal CE | (83 - 33 cal BP)<br>(140 - 88 cal BP)<br>(256 - 225 cal BP) | -21.7 o/oo | High marsh; overlying stratigraphy indicates buried by washaround deposition |
| | **638250** | **organic sediment** | **140 +/- 30 BP** | **(58.2%) 1797 - 1944 cal CE**<br>**(37.2%) 1671 - 1779 cal CE** | **(153 - 6 cal BP)**<br>**(279 - 171 cal BP)** | **-21.7 o/oo** | |
| C7, 65.5-71 cm | 633958 | plant material | 180 +/- 30 BP | (49.9%) 1722 - 1814 cal CE<br>(19.2%) 1656 - 1698 cal CE<br>(19%) 1910 - Post CE 1950<br>(7.3%) 1836 - 1880 cal CE | (228 - 136 cal BP)<br>(294 - 252 cal BP)<br>(40 - Post BP 0)<br>(114 - 70 cal BP) | -22.4 o/oo | Brackish fringe; likely undergoing passive drowning based on up-core δ¹³C |
| | **635979** | **organic sediment** | **270 +/- 30 BP** | **(45.3%) 1618 - 1670 cal CE**<br>**(42.7%) 1508 - 1594 cal CE**<br>**(7%) 1780 - 1798 cal CE**<br>**(0.4%) 1946 - Post CE 1950** | **(332 - 280 cal BP)**<br>**(442 - 356 cal BP)**<br>**(170 - 152 cal BP)**<br>**(4 - Post BP 0)** | **-23.4 o/oo** | |
| C8, 105-111 cm | 633959 | plant material | 130 +/- 30 BP | (64%) 1798 - 1942 cal CE<br>(26.8%) 1674 - 1744 cal CE<br>(4.1%) 1750 - 1765 cal CE<br>(0.5%) 1774 - 1776 cal CE | (152 - 8 cal BP)<br>(276 - 206 cal BP)<br>(200 - 185 cal BP)<br>(176 - 174 cal BP) | -13.2 o/oo | Low marsh; overlying stratigraphy indicates buried by washover |
| | **638251** | **organic sediment** | **280 +/- 30 BP** | **(54.7%) 1504 - 1596 cal CE**<br>**(37.7%) 1616 - 1666 cal CE**<br>**(3%) 1783 - 1795 cal CE** | **(446 - 354 cal BP)**<br>**(334 - 284 cal BP)**<br>**(167 - 155 cal BP)** | **-16.6 o/oo** | |
| C8, 111-118 cm | 633960 | plant material | 180 +/- 30 BP | (49.9%) 1722 - 1814 cal CE<br>(19.2%) 1656 - 1698 cal CE<br>(19%) 1910 - Post CE 1950<br>(7.3%) 1836 - 1880 cal CE | (228 - 136 cal BP)<br>(294 - 252 cal BP)<br>(40 - Post BP 0)<br>(114 - 70 cal BP) | -11.5 o/oo | High marsh; likley undergoing passive drowning based on up-core δ¹³C; plant material may include roots from overlying low marsh or record high to low marsh transition in-situ |
| | **638252** | **organic sediment** | **260 +/- 30 BP** | **(51.5%) 1620 - 1674 cal CE**<br>**(28.7%) 1516 - 1590 cal CE**<br>**(13.6%) 1766 - 1800 cal CE**<br>**(1.6%) 1942 - Post CE 1950** | **(330 - 276 cal BP)**<br>**(434 - 360 cal BP)**<br>**(184 - 150 cal BP)**<br>**(8 - Post BP 0)** | **-21.1 o/oo** | |
| C9, 57-62 cm | 633961 | plant material | 120 +/- 30 BP | (67.2%) 1799 - 1940 cal CE<br>(25.8%) 1680 - 1740 cal CE<br>(2.4%) 1752 - 1764 cal CE | (151 - 10 cal BP)<br>(270 - 210 cal BP)<br>(198 - 186 cal BP) | -16.1 o/oo | High marsh; possibly burying former low marsh, or including plant debris from adjacent low marsh |
| | **638253** | **organic sediment** | **90 +/- 30 BP** | **(69.4%) 1806 - 1926 cal CE**<br>**(26%) 1687 - 1730 cal CE** | **(144 - 24 cal BP)**<br>**(263 - 220 cal BP)** | **-21.9 o/oo** | |




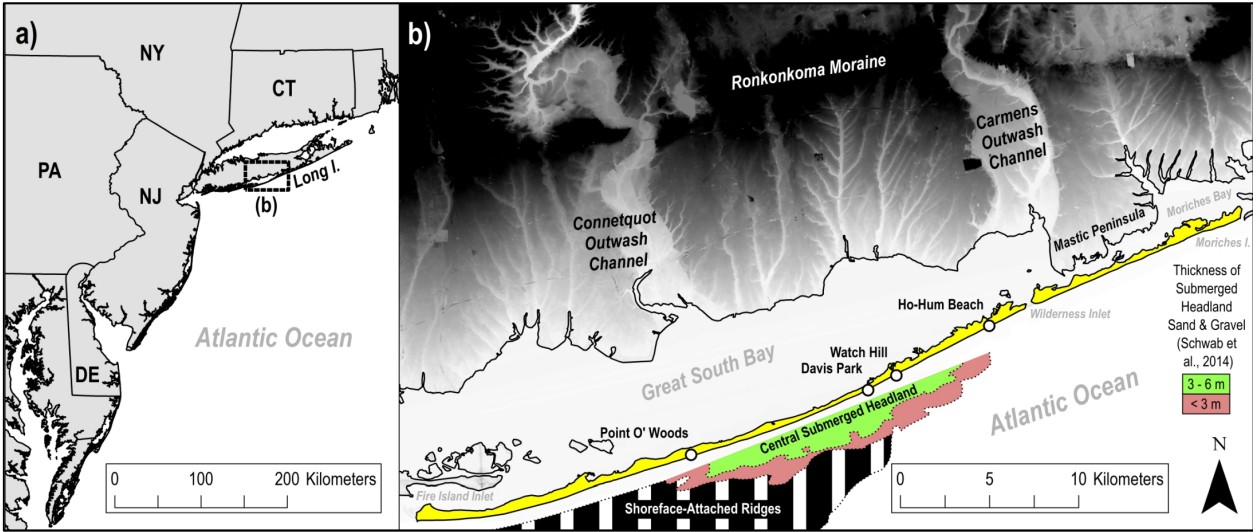

**Figure 1: (a) U.S. Mid-Atlantic region—Fire Island is positioned centrally along the southern coast of Long Island, New York (black box; panel b). (b) Detail of Fire Island, highlighted in yellow, and Long Island, depicted in grayscale lidar-derived topography (white = 0 m, black = 35+ m NAVD88). The two most prominent features of the mainland behind Fire Island are the former Connetquot and Carmens glacial outwash channels, which extend seaward from the Ronkonkoma Moraine. Also shown is an area of shoreface-attached ridges (black and white stripes) and the location and approximate thickness of the Central Submerged Headland (red and green fill)—a shallow Pleistocene sediment lobe that underlies the barrier shoreface between Watch Hill and Point O' Woods. Boundaries and thickness of submerged headland and area of shoreface-attached ridges modified after Schwab et al. (2014). Lidar digital elevation model from FEMA (2006).**





**Figure 2: (a)** Geomorphic zones of Fire Island (modified from Ciarletta et al., 2021), which mirror the east-to-west alongshore transport gradient. **(b)** Transgressive zone (red), which fronts the Mastic Peninsula and has undergone recent and historical erosion and retrogradation. **(c)** Zone of historical (hist.) aggradation and amalgamation (orange), which features 8 m elevation dunes and has undergone recent erosion since the (re)opening of Wilderness Inlet in 2012. **(d)** Zone of prehistoric (preshist.) progradation and amalgamation (yellow), with multiple moderate-relief and shore-parallel dune ridges and relatively stable shoreline positions since the mid-19th century. **(e)** Zone of elongation (green), which has historically elongated westward since at least 1825 and likely earlier. Lidar digital elevation model from Brenner et al. (2016).






**Figure 3: Study sites on/adjacent to central Fire Island, from east to west.** Annotated purple lines are ground-penetrating radar transects with profiles shown in Results. Panel (a) depicts the easternmost limit of the Otis Pike Fire Island High Dune Wilderness, just east of Wilderness Inlet, where the barrier transitions to a historically transgressive single-foredune system heavily modified by human intervention. Panel (b) depicts Ho-Hum Beach, which is in the updrift portion of the central region just west of Wilderness Inlet and features a 5-8 m elevation double dune-ridge system. Panel (c) shows the area around Watch Hill and Davis Park, which is at the interface of Zones II and III, and features an anomalous low spot near cores C6 and C7. Finally, panels (d) and (e) depict a 7-km stretch of barrier centered on Point O' Woods. The eastern 4 km of this section is marked by a succession of relict dune ridges that gently recurve and splay to the northwest at Point O' Woods proper. Lidar digital elevation model from Brenner et al. (2016); base imagery USDA NAIP (2015).





**Figure 4: Morphochronological map of central Fire Island, depicting relationships among the relict and modern landforms of the barrier system; b, c, d at same map scale. Red-hued areas are interpreted as the oldest portions of the barrier platform and comprise two former rotational barrier remnants and an intervening inlet fill—as evidenced by the geometry of relict dune ridges—that predate the historical record. Blue-hued areas are the second-oldest component of the system and correspond with a recurved spit complex and adjacent downdrift inlet at Watch Hill that was historically active at least until 1670 CE. Green-hued areas represent portions of the system that were documented to be historically active from at least the mid-18th century through the early 19th century, while the pale-yellow region corresponds to the extent of the modern foredune, which likely postdates the mid-18th century across most of its length. Base imagery is USDA NAIP (2015).**



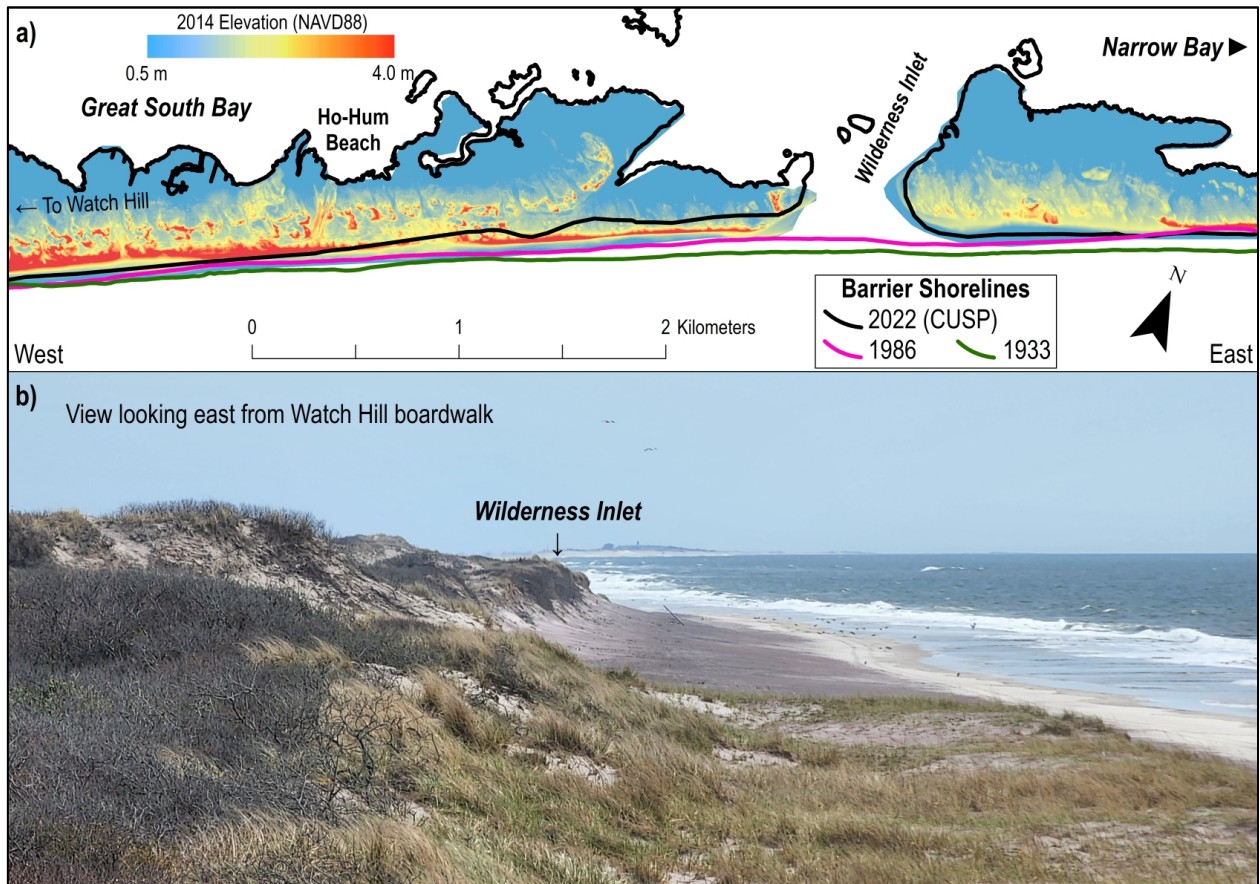

**Figure 5: (a) Changes in shoreline geometry at the eastern end of the Wilderness Barrier, overlain on a lidar DEM from 2014 (see Brenner et al., 2016); 1933 and 1986 shorelines from Himmelstoss et al. (2010). The black line outlines the shoreline of Fire Island in 2022, as represented by NOAA's Continually Updated Shoreline Product (CUSP), which is defined as the MHW shoreline as derived from a variety of sources (see https://shoreline.noaa.gov/data/datasheets/cusp.html). Much of the foredune along the 2.5 kilometers of coast downdrift of the Wilderness Inlet has been eroded. (b) View looking east from Watch Hill towards the Wilderness Inlet in April 2022. A prominent offset has developed between the updrift and downdrift coasts bracketing the inlet. Photo credit: D. Ciarletta, USGS.**

905





**Figure 6:** (a) 2014 Lidar digital elevation model of the Watch Hill-Davis Park area (see Brenner et al., 2014) overlain on 2015 NAIP imagery. (b) Detailed morphologic map of the same, showing structure of relict barrier topography and locations of remnant inlet throats. Dashed red line between cores C6 and C9 (red stars) depicts an inferred shoreline probably dating to the early 19th century based on comparison with U.S. Coast and Geodetic Survey Chart H-46, 1835. "Truncated Beach" and "The Lobe" refer to updrift and downdrift complexes of recurved and washaround ridges inferred to exist on distinct depositional platforms within the zone of spit elongation. Their alongshore subsurface structure is shown in Figure 8. (c) Pre-development Beach Erosion Board (U.S. Army Corps) aerial imagery of Davis Park-Watch Hill area, with inset (d) showing presence of washaround ridges ringing a central high point on an abandoned spit deposit—"The Lobe"—updrift of Whalers Inlet (Modern aerial: USDA NAIP, 2015).







**Figure 7: Cross-shore uninterpreted (a, c) and interpreted (b, d) GPR profiles of the same line at Ho-Hum beach in 2016 (a, b) and 2021 (c, d), showing the structure and lithology of the Wilderness Barrier near the eastern end of Zone II. Seaward-dipping reflections south of Core C4 (panels c, d; yellow) are consistent with past progradation, while landward-dipping reflections to the north highlight relict washover. A gently undulating reflector (magenta) under the landward-dipping units at approximately -3 m elevation may correspond with the antecedent Pleistocene surface. Modern foredune visible at the seaward end of 2016 profile has been destroyed by erosion in 2021, leaving behind only a low, transgressive remnant about 60 m landward of the original dune position.**





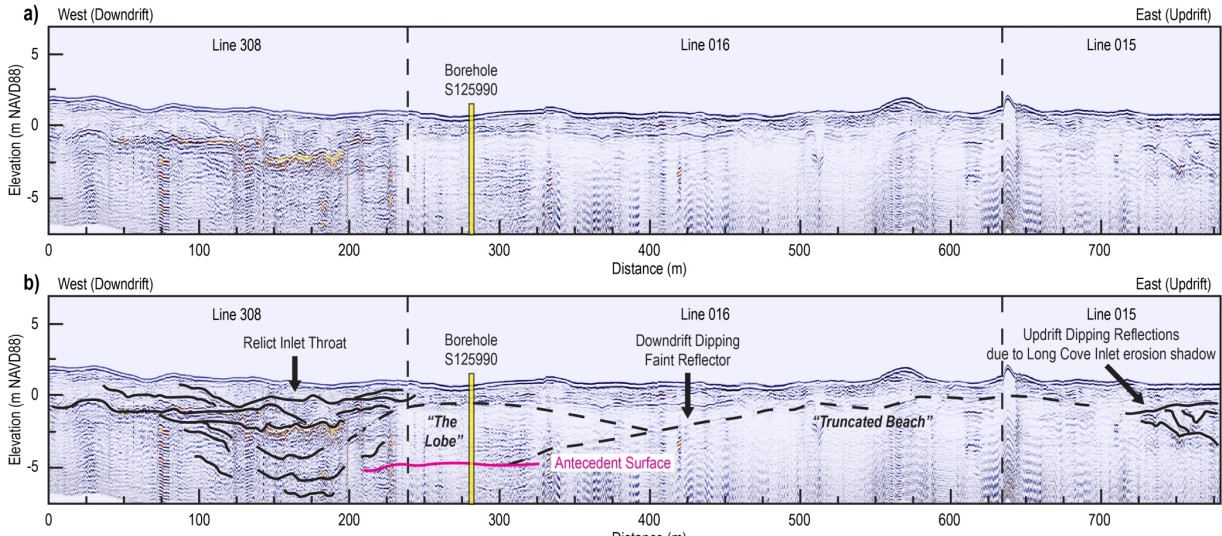

**Figure 8: Alongshore uninterpreted (a) and interpreted (b) GPR transect behind the modern foredune at Watch Hill (east/updrift) and the former Whalers Inlet (west/downdrift). Scans reveal the structure of the updrift platform of the spit complex, "The Lobe,"**
**upon which cores C6 and C7 are sited, and a relict inlet throat that probably represents the final position of Whalers Inlet. The inlet cuts to an elevation of at least -6 m NAVD88, which is below the depth of the observed transgressive surface underlying the barrier at this location.**

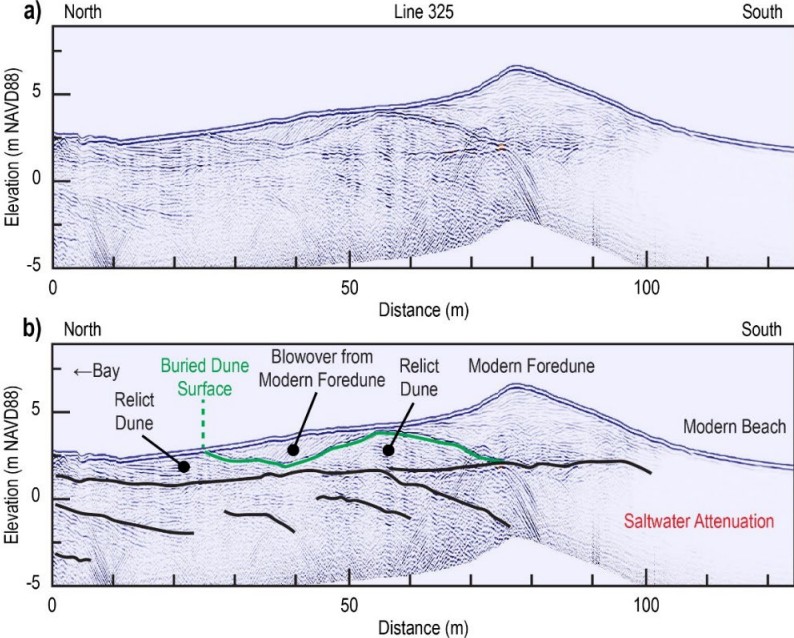

**Figure 9: Shore-perpendicular uninterpreted (a) and interpreted (b) GPR transect west of Davis Park, slightly more than 2 km downdrift of former Whalers Inlet. The profile reveals seaward-dipping reflectors consistent with past progradation of the Barrett Beach Barrier. Above these reflections, the active foredune has amalgamated against a relict dune line and infilled a relict swale with blowover.**



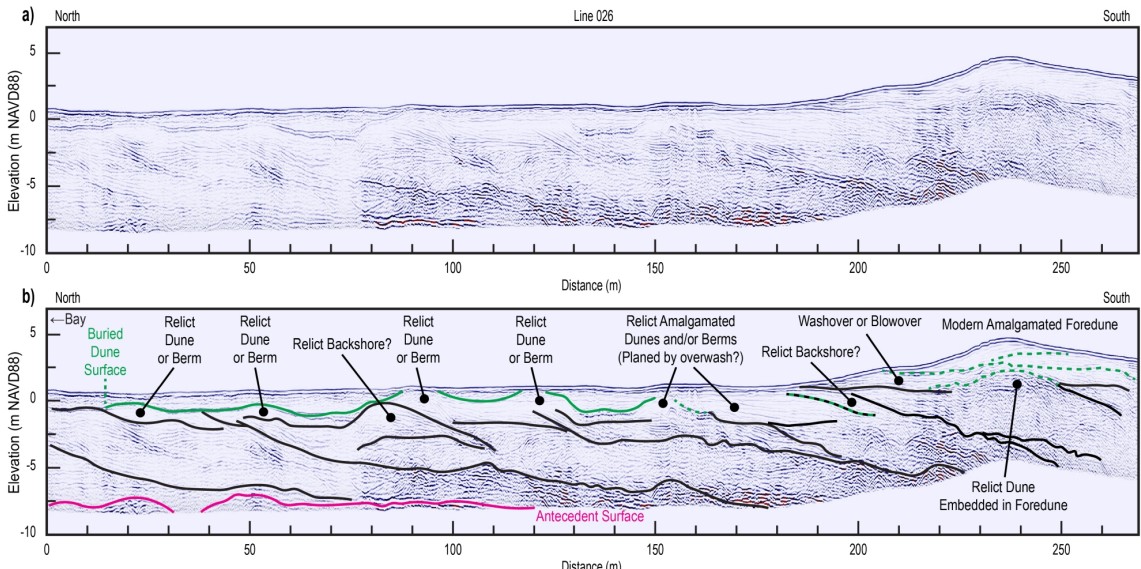

**Figure 10: Shore-perpendicular uninterpreted (a) and interpreted (b) GPR transect just east of Point O' Woods, near the Sunken Forest. The profile depicts a 250+ m-wide succession of progradational clinoforms overlying a possible antecedent surface (magenta) at a depth of -7.5 m NAVD88. Low-relief relict dune ridges and possible relict berms are also present in the subsurface behind the landward limit of the modern foredune. Buried dune surfaces are indicated in green, with dashed green lines indicating dune surfaces buried by aeolian amalgamation. The black-and-green dashed line indicates a dune surface buried by possible beach facies.**

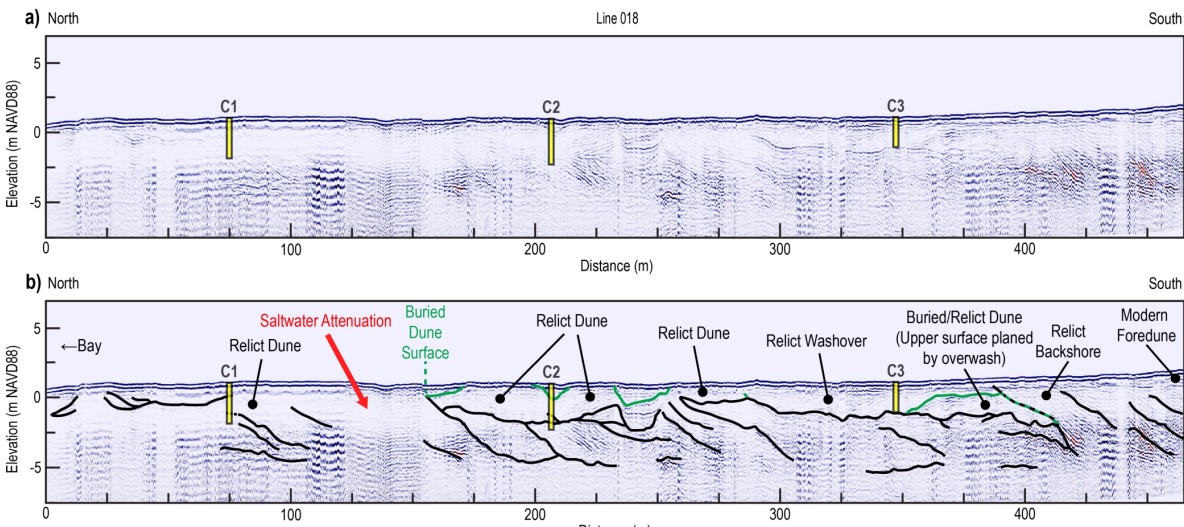

**Figure 11: Cross-shore uninterpreted (a) and interpreted (b) GPR profile at Point O' Woods showing location of sediment vibracores and interpreted stratigraphy based on correlation with cores. The stratigraphic structure of the transect is primarily progradational and features a succession of partly buried relict foredunes. Additionally, there is a possible transgressional dune preserved in the subsurface at ~380 m seaward of the lagoon shoreline. Washover overlies the dune and onlaps an interpreted relict swale surface in the landward direction. In the seaward direction, progradational beach clinoforms offlap from the relict dune surface and underlie the heel of the modern foredune. Buried dune surfaces are indicated in green, and the black-and-green dashed line indicates the dune surface buried by possible beach facies.**



**Figure 12: Uninterpreted (a, c) and interpreted (b, d) GPR transects of intersecting north-south line 20 and east-west line 275, located 2 km downdrift of Point O' Woods (see Figure 3e for location). Profiles depict steeply dipping reflections in both the seaward (c/d) and downdrift directions (a/b), consistent with spit development.**





**Figure 13: Relative elevations of sediment cores (NAVD88) and interpretation of lithologies. Green dots mark locations of radiocarbon samples, with ages reported for both plant material and organic sediment. Modern mean high water (MHW) is indicated at +0.46 m.**



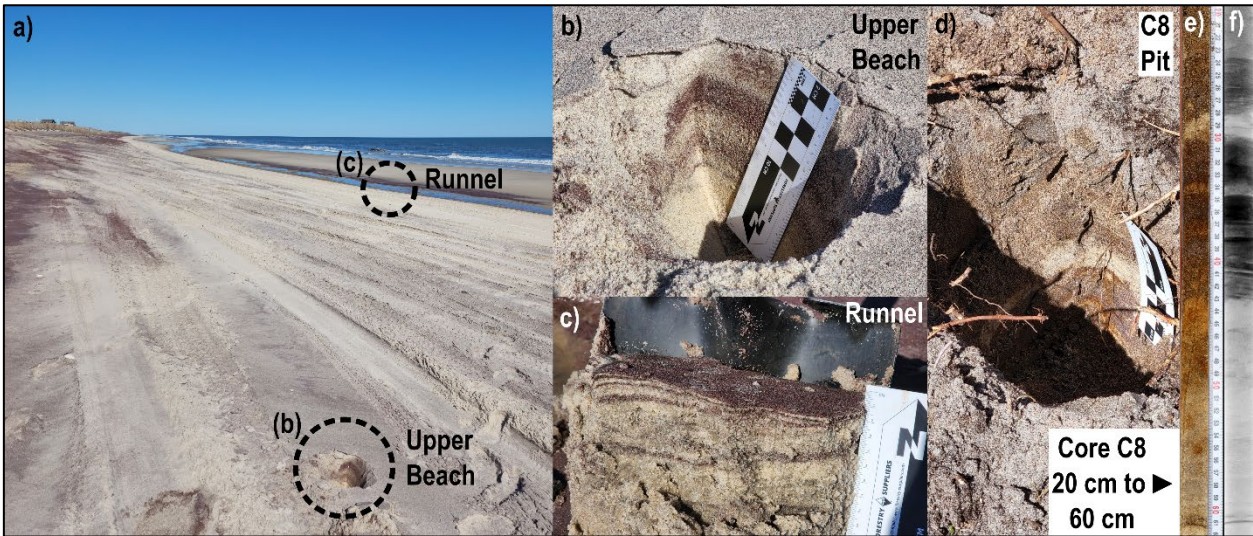

**Figure 14: Visual comparison of core C8 with post-storm beach lithology. (a) Overview of beach at Davis Park two days after the April 2022 nor'easter. (b) Small pit dug into modern upper beach, revealing alternating layers of heavy mineral and clean sand. (c) Dug-out section of runnel floor revealing thin, deformed heavy mineral and clean sand bands. (d) Pit at site C8, revealing in-situ lithology comparable to modern beach. (e) C8 core section between 20 and 60 cm, showing detailed lithology. Some orange stain is apparent from groundwater. (f) X-ray image of section shown in (e). Darker units contain heavy minerals. Photo credit (a-d): D. Ciarletta, USGS.**



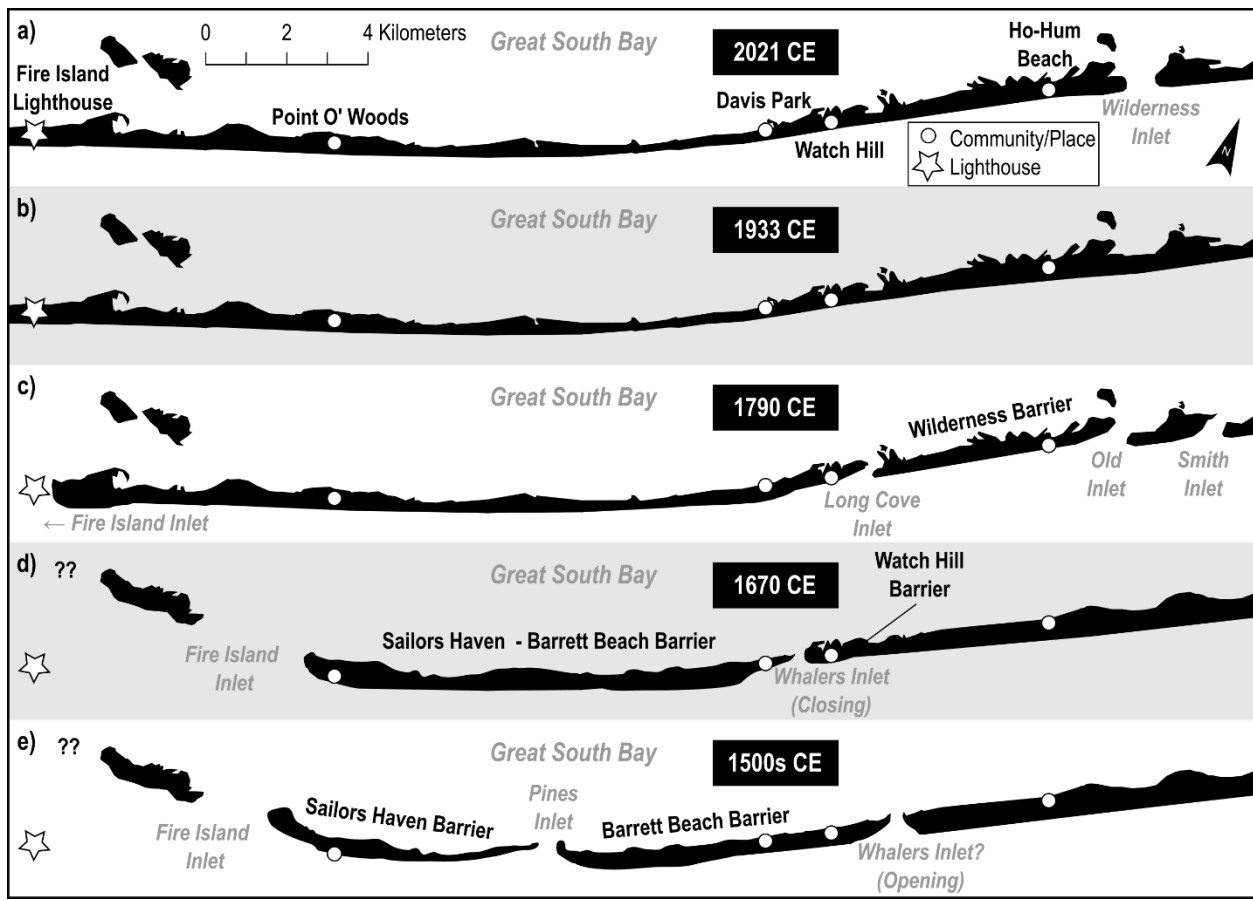

**Figure 15: Timeline of central Fire Island's evolution from the 16th century to the present, based on a synthesis of geomorphic interpretation and age control from this study, as well as historical accounts and analyses described in previous studies and reports (Clark, 1986; Leatherman and Allen, 1985; McCormick, 1984; Ruhfel, 1971; Suydam, 1942).**