# Peer review of "Implications for the Resilience of Modern Coastal Systems Derived from Mesoscale Barrier Dynamics at Fire Island, New York"

_EGUsphere, 2023_

## Author Response (AR1)

**Review #1 Anonymous**

**[[ This manuscript reconstructs the recent evolution of Fire Island barrier islands, determining the main processes responsible for their past (century scale) evolution, with the objective of demonstrating how managed barrier islands evolve in a totally different manner than natural barriers to finally discuss the current and future implications to barrier resilience or barrier capability to adapt to sea-level rise. The present work shows a good example of how human strategies applied to dynamic coastal systems may impact provoking shifts in barrier morphological state and even changing the main controlling processes. Despite of the interest of this topic, I find it difficult to understand what is new from this work, not only in relation to the concepts or ideas, but also in relation to the data and the actual temporal reconstruction. According with the cited references a good amount of work has been already done in this area, which already stated the importance of the inlets to their evolution as well as of human management activities. In fact, previous works have already discussed the "problem" of managed barrier islands in relation to morphological adaptation treating them as coupled Human-Landscape systems (e.g., McNamara and Lazarus, 2018). ]]**

+ We are happy to see that the work convincingly demonstrates that Fire Island is a heavily altered system. Although there is evidence of this at Fire Island in existing literature, it has not been synthesized (or combined with modern data) in a way that conclusively shows this until now. We note that one of the major reconstructions of Fire Island's evolution by Clark (1986) suggested Fire Island was actually relatively stable prior to the late 18th and 19th centuries. We demonstrate that this is not true—Clark's dataset focused on the eastern half of the island, while we fill in the study gap that has historically existed in the west, determining the timing and nature of morphologic changes suspected in but not clearly supported by previous studies. In terms of implications, although the "problem" of managed barriers in relation to morphological adaptation is not new (in the U.S., this stretches back to the 1970s with Dolan et al.'s studies in the Outer Banks), there remain emerging concepts to explore. Most notable is the concept of "geomorphic capital" and its effects on both short- and long-term resilience (Mariotti and Hein, 2022). In this respect, not a lot of work has been done to show where geomorphic capital is important, or to quantify its effects—to explore the latter, we must first explore the former.

**[[ In this regard, I would recommend the authors to more clearly state what is new from their work, both in terms of data and ideas. Regarding the second, the authors made a big effort to discuss the implications of this human induced steady-state, however, this is not a new idea, so I would recommend to better frame their point. In fact, in subsection 5.3 known facts have a greater weight than the unknown ones. After reading it, one gets the idea that the authors already know how the island is going to respond, by eroding the dune and drowning the backbarrier, so it would help if the unknowns are more clearly presented. I wonder if this expected evolution will result even if human attempts to hold the line continue over time or do the authors expect that after investing on managing the islands the strategy will radically shift? Did the authors consider the possible contribution of "blowover" sands to maintain the island platform? Figure 9 shows how this process may have had an impact in the recent past. I also find interesting the idea that a high dune adds short-term resistance**

**(one of the dimensions of resilience following Kombiadou et al. 2019) but will enhance retreat in the long-term. Could the authors extend a bit further on that in terms of resilience? ]]**

+ It is apparent from this section of the reviewer's comments that some of the concepts we go through in the paper were not clear enough, so we make some changes to emphasize these points throughout. Specifically, we reiterate in the Introduction that we collect new data *and* synthesize existing science to form a complete picture of Fire Island's evolution, with a focus on the central-western part of the island, where age control and stratigraphy were lacking. Additionally, we explicitly point out the importance of this type of study in evaluating the impacts of geomorphic capital.

Regarding the unknowns, we now emphasize that while we can reasonably guess how the island could evolve, there is a lot we do not know, specifically in terms of how fast these changes could progress, or if it is possible to focus management efforts to mitigate them (at least over the short term). We reiterate that one of the purposes of this study is to highlight the particulars of Fire Island's present state in the context of past morphologic change, providing a basis for forward modeling.

Addressing the discussion of resilience, the contribution of "blowover" would absolutely have an impact on geomorphic capital, which in turn would alter both the short- and long-term resilience of the island. We would argue that the blowover constitutes the rear flank of the dune and is indicative of the amalgamation of the large primary foredune with relict ridges in the rear. Thus, this would be synonymous with the large, aggrading foredune that we note throughout the modern system. In the results, we adjust the description of the amalgamation to make this point more clearly: "The modern foredune is amalgamated against the most seaward relict ridge and partly amalgamated with the second line of relict dunes through blowover deposition." We agree that, in terms of dimensions of resilience, this can be explained more clearly, and we make some modifications to reference definitions in Kombiadou et al. and another work suggested by the second reviewer (see: Masselink and Lazarus, 2019).

***Specific, significant updates to text***

**1st paragraph of Introduction extended to include a primer on resilience:**

"Here, we describe morphologic resilience as the capacity of the coastal landscape to maintain the distribution and character of its subaerial ecomorphological features through time (Masselink and Lazarus, 2019), with state shifts comprising threshold changes in system morphology that cannot be easily recovered to a previous configuration (Kombiadou et al., 2019)."

**3rd paragraph of Discussion Section 5.3 also modified to include a reference to Kombiadou et al., (2019):**

"Loss of morphological resilience is likely to be exacerbated by reserves of naturally available sediment in reaches of relict progradational dunes, which could enhance Fire Island's short-term resistance (see Kombiadou et al. 2019) against landward migration but ultimately increase the

island's long-term committed retreat (Mariotti and Hein, 2022) and persistence in an eventual low-elevation/transgressive state."

**The last paragraph of the Background is modified and extended as follows to emphasize resilience and geomorphic capital:**

"Although reliable records indicate long-term barrier stability in central Fire Island, particularly in Zone III, other authors suspected this section of the barrier was much more dynamic prior to extant historical observations (Leatherman and Allen, 1985; McCormick et al., 1984). Regardless of the exact nature and temporal framework of its stability, the relative longevity of this section of the barrier (Figure 2c/d; Zones II and III) suggests it contains a long-term, prehistoric geomorphological record of past changes in sediment fluxes and environmental forcing that could be used to develop a baseline of Fire Island's natural morphologic variability and resilience. Based on this assessment, we collected comprehensive geologic data from Zones II and III. Along with similar data from adjacent areas of Zones I and IV, and combining our new data with past studies and historical observations, we (1) determine the timing of relict ridge and beach formation in Zones II and III, (2) identify the drivers of changes in sediment availability that influenced island evolution, and (3) infer possible evolutionary pathways for Fire Island and other barriers in the future. On the last point, the presence of relict ridge successions in central Fire Island also compels us to consider the impacts that geomorphic capital might have in the future, especially since this is an emerging area of concern that has been poorly described and quantified until recently (Hein and Mariotti, 2022)."

**Finally, the 3rd paragraph of Introduction is also revised to emphasize the purpose of the study:**

"Here, we use geomorphic mapping of active and remnant dune features at Fire Island, a semi-natural barrier island in New York, USA, to gain insight into both natural and anthropogenic drivers of barrier landscape change. Although Fire Island has been the subject of numerous field investigations and modeling studies (Leatherman, 1985; Lentz and Hapke, 2011; Locker et al., 2017; Schmelz and Psuty, 2022; Schwab et al., 2000; Schwab et al., 2014; Zeigler et al., 2022), little is known about the island's internal structure or the timing of its development, especially in its central ~24 km. This presents a variety of issues for forward modeling and management practices. Without a baseline of past morphologic variability, it is not clear what the natural character and distribution of ecologies in the Fire Island system were prior to significant human interference in the landscape, or what the most important drivers were in shaping the barrier. Subsequently, it is not known how resilient the system was in the past, and whether the current system reflects a significant morphologic state shift from a previous configuration or is currently in transition to a new state. To fill the knowledge gap, we synthesize existing studies with historical documentation of the island's landscape and new geomorphic mapping to reveal locations where significant records of morphologic change are preserved. Ground-penetrating radar investigations coupled with coring and radiocarbon dating then provide chronological control and paleoenvironmental information. Finally, combining this information, we reconstruct the evolution of central Fire Island to understand differences between present and past morphologic states, including how such differences could affect the system's future resilience and the implications for mesoscale behavior of barrier systems globally."

**[[ Regarding data, maybe include previous works only in the study area description and discussion. This would reduce the length of the results section and likely make it less descriptive. I would also recommend the authors to summarise the results from the dating of so many samples and how they actually help to constrain or better built a chronology of the area. In this line, I also recommend reorganising some of the results in this line, Figure 4 seems a chronological reconstruction rather than morphological, in fact the only morphological elements are the lines stating the ridges and the likely position of inlets. If this chronology uses the new dates, I would present it later and maybe to support of summarised information from the dated samples. A morphological map of Watch Hill-Davis Park area (Figure 6) is missing for the entire study area. This could include not only the difference between relict and modern dunes, but maybe additional information on elevation as some of the relict dunes seem to reach similar elevations as the modern dune (Figure 6). How do the authors explain those high dunes? Are these common features? ]]**

+ Based on comments from another reviewer, we relabel the Results as "Results and Interpretation" and briefly discuss this format in the Introduction. To make the radiocarbon results easier to follow and compare with dates shown in Figures 4 and 15, we convert the conventional radiocarbon ages to calendar years in both the text and Table 1. We note that Figure 4 is indeed a 'morphochronological' reconstruction, and we now make it more clear in the text and caption that the dates provided in that figure are based on a synthesis of existing observations (we go over our age control from cores later in the Results).

A morphological map of the entire study area would be a considerable effort to produce and would create more results for an already very long results section. Such a level of detail is not needed outside of the Davis Park-Watch Hill area since the morphology elsewhere is relatively less complicated.

Regarding dune heights, we realize that figure 6 might give the impression that relict dunes are as tall as the modern foredune, but for the most part, they are much smaller. Our elevation scale only goes to 2 m specifically to bring out the details of the relict dunes. In reality, the true elevation of the modern foredune exceeds 4 meters. We update the scale and caption of figure 6 to make this clearer.

***Specific, significant updates to text***

**We modify the first paragraph of the Method Section 3.1 to mention that we use observation/mapping records from McCormick et al. (1984) and Strong (2018) in our morphochronological interpretation:**

"We first assessed central Fire Island's geomorphology using lidar digital elevation models (DEMs) from 2020 (U.S. Army Corps, 2021) and 2014 (Brenner et al., 2016), with additional information about long-term geomorphic change derived from historical shorelines (see Allen et al., 2002; Himmelstoss et al., 2010; and Terrano et al., 2020) and other observational/mapping records (McCormick et al., 1984; Strong, 2018)."

**We specifically call out Figure 4 as a morphochronological map in the first sentence of Results Section 4.1:**

"Relict dune ridge structure in central Fire Island reveals an updrift-downdrift morphologic dichotomy (see Figure 4 morphochronological map; compare b with c/d)."

**We amend the first sentence of the Figure 4 caption to explicitly state where chronological control comes from:**

"[…] with dates from historical observations (McCormick et al., 1984; Strong, 2018."

**[[ Regarding data interpretation, I suggest the authors to revisit some of their interpretations from the GPR or explain better their interpretations, which I assume were done by the authors. Were those based on modern analogues? If so, which is the typical elevation of the transition between the dune and the beach? If the intention of the authors when using the geophysical data is only to identify the direction of growth, then maybe I would avoid including different type of environments as they might not be possible in some cases. In this regard, I also missed some justification for the attribution of depositional environments to the sediment cores. How the authors explain the accumulation of aeolian sediments close to -2m when the GPR data and other cores seem to suggest that the transition between the dune and the beach is around 0? What do the black lines in the interpreted panels of the radargram mean? It is a bit confusing as the typical interpretation based on facies was not applied here. ]]**

+ It appears there is confusion surrounding where the transition between the dune and beach should be in the GPR profiles based on an absolute elevation relationship among cores. The reality is that the elevation of the dune-beach interface has changed through time due to increasing sea level. For example, in Core C2 we observe the dune-beach transition at around a meter and half below NAVD88 because these sediments were deposited more than 763 calendar years ago (690 years BP recorded in C1), at a time when sea level was lower. The present rate of rise around Long Island is about 3 mm/yr, but even using a slower rate of rise of 2 mm/yr would yield 1.6 meters of sea level change over 800 years. We now explicitly call this out in the text. "The age of 690 yrs BP at C3 implies that relict beaches and ridges landward of this location (e.g., at C1 and C2) are even older. This interpretation is supported by the elevation of the C1/C2 beach-dune interface near -1.5 m, which demonstrates that aeolian deposition occurred at a time when sea level was lower than at present."

Regarding the black lines in the radargrams, these merely highlight prominent reflections in the presumed island platform (washover, beach, and inlet facies) and serve to inform the reader of the orientation of strata. Captions have been amended to read as follows: "Black lines highlight prominent reflections within and between presumed washover, beach, and inlet facies."

Our interpretation of environment (where possible) is based on both observations of surface morphology and lithology from cores, the latter of which is available through the provided assets; see Data Availability. We do take more of an overview-level approach to identify environments by GPR alone, which is why, for example, we do not label possible relict surge channels, landward

migrating bars, or other more nuanced features that were not represented in our coring efforts or directly coupled with surface features.

**[[ Finally, I find the conclusions are based to a great extent on the discussion and thus not sustained by the results from the present work, in particular when the authors suggest that the barrier is approaching a geomorphic tipping point. This was not introduced before in the text and would need further explanation. In the same section they also mention in line 580 "bay erosion", which was not discussed along the present work. The same applies to the title, which focuses on the unknown implications of the human-induced steady-state rather than what can be drawn from the data. ]]**

Objectively, half of the sentences in the Conclusions focus on summarizing the results alone, so the issue is that the takeaways in the latter part of this section are not contextualized as well as we would like elsewhere in the manuscript. Based on this review, we have determined this has mostly to do with our Introduction and Background, which have now been adjusted to steer the reader towards our key points upfront. Specifically, we emphasize that, even with the wealth of knowledge surrounding Fire Island, we do not know how dynamic it was prior to human interference, which means it is hard for us to calculate the effect of geomorphic capital on system evolution or understand how the system should be working in its natural condition. Both of these are significant to any potential management actions to alter the island's ecology or physical resilience.

***See key points from our reworked Introduction and Background paragraphs, specifically…

**…from final paragraph of Introduction:**

"Without a baseline of past morphologic variability, it is not clear what the natural character and distribution of ecologies in the Fire Island system were prior to significant human interference in the landscape, or what the most important drivers were in shaping the barrier. Subsequently, it is not known how resilient the system was in the past, and whether the current system reflects a significant morphologic state shift from a previous configuration or is currently in transition to a new state. To fill the knowledge gap, we synthesize existing studies with historical documentation of the island's landscape and new geomorphic mapping to reveal locations where significant records of morphologic change are preserved."

**…and the last paragraph of the Background:**

"[…] the presence of relict ridge successions in central Fire Island also compels us to consider the impacts that geomorphic capital might have in the future, especially since this is an emerging area of concern that has been poorly described and quantified until recently (Hein and Mariotti, 2022)."

**[[ Line 90 says Carmans River while in Figure 1 is written as Carmens ]]**

+ Updated. Thanks!

**[[ Line 408, I believe the it after "C2," is misplaced ]]**

+ Correct. The "it" should come after "but."

**[[ Line 478 the inlets were and not "was" naturally formed ]]**

+ The "was" is referring to Moriches Inlet in the singular (not including Wilderness Inlet). The interjection here does not add anything, however, and has been deleted.

Review #2 Anonymous

**[[ I read with interest the contribution by Ciarletta et al. on "Implications for the Resilience of Modern Coastal Systems Derived from Mesoscale Barrier Dynamics at Fire Island, New York". This contribution fits within the scope of Earth Surface Dynamics as it describes the morphostratigraphic changes that have taken place along the Fire Island barrier chain over the past ~500 years and builds on a large baseline of coastal barrier research along the US east coast and around the world, as well as previous studies of this particular area.**

**The results of the study comprise interpretation of LiDAR-derrived DEMs, Ground Penetrating Radar (GPR), sediment cores and a new set of radiocarbon dates. The dataset is robust and builds upon several other studies, adding new information for a comparatively less studied portion of this coastal barrier. The new results are integrated with past studies to produce an overall picture of coastal barrier deposition over the past 500 years, summarised in the final figure of the manuscript – Figure 15. The methodology is sound and due credit is given to previous work. ]]**

+ We are glad the manuscript held the reviewer's interest and appreciate the acknowledgement of the extensive research we put into crediting previous work. There has been a great deal of scientific interest in Fire Island by various authors over the years, but aside from the works of Leatherman and Allen decades ago, there has been little effort to synthesize studies and investigate some of the ideas of island evolution put forward 30-40 years ago.

**[[ The title and abstract are adequate and the overall presentation of the data is reasonably clear, although some improvements are suggested to the figures below. The prose is well written and the article follows a logical structure. However, I suggest that the Results section may be better described as Results and Interpretation as much of what is presented here fall under the latter category. In fact, overall the results are very long and maybe there is an opportunity to trim these down by including some of the more exhaustive descriptive sections in supplementary material. I also wonder whether all the GPR figures, which are of high quality, are really needed in the main body of the article? I think the main points could still be communicated with some of these in the supplementary material. Or perhaps presenting only the interpreted sections as one or two combined figures and leaving the combination of uninterpreted and interpreted versions for the supplementary information. My guess is that many readers will be comfortable with the interpreted versions and those with a particular interest in this technique can pursue the supplementary figures. ]]**

+ We agree that the Results are more accurately described as "Results and Interpretation" and update this section accordingly. We also concur with the assessment that the descriptions are long, and we make considerable effort to trim these down and use more precise language throughout.

We appreciate the suggestions regarding the GPR figures; however, our preference is that supplements and supporting materials should not contain any interpreted figures that are also primary results. To conserve length, we had also considered attempting to merge some of the panels from different figures, but this comes at a cost of resolution and mismatched distance scales; it may also confuse the reader due to profiles from different geomorphic zones being shown in the same figure. Consequently, given that eSurf provides the format needed to support the number and scale of figures put forward, we choose to maintain the current order of GPR figures in this section. Additionally, while some readers might be comfortable with just the interpreted GPR figures, we choose to show both uninterpreted and interpreted panels, similar to has been done in existing literature (see Shulmeister et al., 2016 [published with Copernicus] / https://doi.org/10.5194/cp-12-1435-2016; Oliver et al., 2017 / https://doi.org/10.1016/j.margeo.2017.02.014; Nielsen et al., 2009 / https://doi.org/10.1016/j.jappgeo.2009.01.002; and Seminack & McBride, 2019 / https://doi.org/10.2110/jsr.2019.14).

**[[ One other general comment needs to be made. In relation to the radiocarbon dating results, have the author's considered the offset in years between historical dates from say old maps and years BP where 'present' in BP is 1960. There is an inherent ~70 year offset between BP and the present day. This struck me when I noted that many of the radiocarbon ages are quite young — less than 200 years. ]]**

+ This offset is accounted for. Almost all the very young dates (e.g., 30, 120, 130, 140 yrs BP) are plant material intruding older units. The sole exception to this paradigm is the radiocarbon sample taken from the bottom of Core C9, where plants and organic sediment overlap in age. In that case, the conventional radiocarbon ages (120 to 90 yrs BP) would equate to the timeframe of 1830 to 1860 CE based on the 'present' year of 1950. Considering this comment and a similar comment from another reviewer, we include the conversion to conventional calendar years CE in the Results (including Table 1) to avoid confusion.

**[[ Line 7: I prefer "rates of sea-level rise" here ]]**

+ Agree, and updated.

**[[ Line 26: I suggest the recent chapter on barriers led by McBride et al. might be best here. https://doi.org/10.1016/B978-0-12-818234-5.00153-X ]]**

+ Thanks for reference. Will add that here.

**[[ Line 29: A definition of morphologic resilience is needed, and maybe the concept of resilience overall. Maybe look at this paper: Masselink, G., Lazarus, E.D., 2019. Defining coastal resilience. Water (Switzerland) 11, 1–21. https://doi.org/10.3390/w11122587 ]]**

+ The referred paper has overlap with some of the concepts put forward by Kombiadou et al. (suggested by the other reviewer), and we now use these two to briefly define the concept of resilience early in the paper:

"Here, we describe morphologic resilience as the capacity of the coastal landscape to maintain the distribution and character of its subaerial ecomorphological features through time (Masselink and Lazarus, 2019), with state shifts comprising threshold changes in system morphology that cannot be easily recovered to a previous configuration (Kombiadou et al., 2019)."

**[[ Line 29-30: A general reference for rising sea-levels and increasing storm frequency and intensity is needed here. ]]**

+ Added reference to IPCC 6$^{th}$ Assessment. See: https://doi.org/10.1017/9781009157896.013

**[[ Line 31: Is 'morphologic change' here a 'state' change as per the term 'state shift' in the abstract? Define. ]]**

+ Changed to "morphologic state shifts" to be consistent with previous terminology in text.

**[[ Line 35: 'Significatly controlled' is an awkward phrase and 'significant' should be reserved for statistical purposes IMO. ]]**

+ Changed to "primarily."

**[[ Line 36: 'These drivers are a function of' change to 'These drivers are in turn a function of' ]]**

+ We concur and update accordingly.

**[[ Line 38: A different expression is needed here: "inherited morphology (relict barrier morphological features" is awkward and uses 'morphology' twice. ]]**

+ Changed to "relict or 'inherited' morphology."

**[[ Line 46: Again here, is it a morphologic state change? ]]**

+ It is. Text updated to "morphologic state change."

**[[ Line 52: "and recent attention has been placed on their importance" is an awkward phrase – revise. ]]**

+ Revised to "…which have been used to illuminate past evolution…"

**[[ Line 61: replace 'identifies' with 'reveals' ]]**

+ Agree, and updated.

**[[ Line 63: replace 'In total' with 'Overall' ]]**

+ Agree, and updated.

**[[ Line 64: Again morphologic states needs more attention in the Introduction in my view. Especially as the concept seems key to the abstract. What exactly are these states and where is the idea and term from? ]]**

+ We add a line in the Introduction to set up this concept, drawing on Robbins et al. (2022) to describe the key barrier behaviors at decadal scales (e.g., progradation, elongation, narrowing, shortening, migration, rotation) and their accompanying morphologies:

"The net morphological effects of mesoscale barrier dynamics are manifested through several key barrier behaviors. We define these behaviors according to Robbins et al. (2022), including concepts such as seaward/alongshore growth (progradation/elongation), cross-shore/alongshore erosion (narrowing/shortening), as well as differential erosion and progradation (barrier rotation)."

**[[ Line 99: Needs a ref at the end of the sentence ending "littoral sediment supply." ]]**

+ Added reference to Leatherman and Allen (1985) and Kana (1995).

**[[ Line 117: 'Elongational' is an unusual term. Is this common in US east coast barrier literature? ]]**

+ See comment for line 64; this is a term used commonly on the US east coast.

**[[ Line 118: Foredune 'arcs' – again, not sure if this is the correct term – recurved foredunes? Or recurved foredune ridges? ]]**

+ "Recurved foredune ridges" would be the correct terminology; implemented.

**[[ Line 161: is horizontal resolution the resolution of the grid? I.e. Grid cell size? Clarify. ]]**

+ Correct. Clarified as "grid resolution."

**[[ Line 231: Isn't this too early for a reference to Figure 15! ]]**

+ Agree. Reference to Figure 15 removed to be consistent with the rest of the figure references in this section.

**[[ Line 276: 'support' instead of 'confirm' ]]**

+ Changed to "support."

**[[ Line 279: 'steep' – what are the angles? Or annotate them on the Figure. 'seaward-dipping' is preferred to 'ocean-dipping' IMO. ]]**

+ Changed to "relatively steep seaward-dipping"; this is in comparison to the landward-dipping reflections.

**[[ Line 282: 'formally transgressive' – but included elements of progradation ]]**

+ Progradation occurred after the transgression. Revised to make this clearer:

"These observations imply that the updrift portion of the Wilderness Barrier was transgressive prior to the episode of progradation inferred between 1834 and 1933."

**[[ Line 291: 'washaround' ridges? What are these? Is this the correct term? Again in line 373. ]]**

+ This is the correct term. 'Washarounds' were defined by Price (1958); reference added to text. See: https://archives.datapages.com/data/gcags/data/008/008001/0041.htm

**[[ Line 376: 'mud balls' – what are these? Is this the correct term? ]]**

+ This is also the correct term. Mud balls are mud clasts that have been rolled into a spherical shape due to the presence of tides/currents. Mukhopadhyay et al. (2022) summarize the environments in which they occur; reference added to text. See: https://doi.org/10.1002/gj.4310

**[[ Line 440: 'confirming' change to 'supporting', add comma after 'of' ]]**

+ This sentence was generally clunky. We simplified it to read as follows:

"By seeking to confirm impressions of past barrier behavior, we can move beyond local characterization and explore how inherited and modern morphology might impact the resilience and evolution of the island in the future."

**[[ Line 443: change 'part' to 'role', use different word instead of 'fate' ]]**

+ Changed to "role"; instead of fate, we revise the last part of the sentence to "…as well as what role human interventions play in modifying these systems."

**[[ Line 447: 'geomorphic complexity in barrier systems' – maybe this is better described as 'barrier system behaviour' ]]**

+ Changed to "…are a primary driver of barrier behavior and geomorphic complexity."

**[[ Figure 1: Mark the direction of longshore drift here, or maybe on Figure 2a. ]]**

+ Done. Added to Figure 2a.

**[[ Figure 2: I think the coloured lines outlining each barrier could be thinner, or maybe not needed on panels b-e if b-e are labelled on panel a. Also a hillshade layer with transparent colours might be a better way of showing the LiDAR data. There is also a lot of white space in each panel and not much DEM – is there some way of making these bigger? – like those of Figure 3. ]]**

+ Made colored lines thinner. It is not possible to make the sections bigger without adding more panels (the island is very thin) or greatly expanding the width of the figure. We do, however, provide the figure in fairly high resolution so that it is possible to zoom in a see additional detail. This is also the reason why we created Figure 3; so that the reader could get a detailed view of the study area morphologies, which are representative of the four geomorphic zones.

**[[ Figure 6 is the best of the first 6 figures. I suggest having a look a the figures and portrayal of data in Kennedy et al. (2020) for some inspiration if needed. https://doi.org/10.1016/j.margeo.2020.106366 ]]**

+ Thanks! The figures in the provided reference are also excellent, although one thing we should mention is that we deliberately picked more simplified color schemes for terrain shading to make it easier on individuals with color vision deficiencies. This is also the reason why we use zone labels in the sidebar on Figure 2 and use checkered/uncheckered GPR lines to represent profiles not-shown/shown in figures. With the exception of the middle panel in Figure 6, the rest of our figures function reasonably well in colorblind simulation (see: https://www.color-blindness.com/coblis-color-blindness-simulator/).

**[[ Figure 15 as the culminating figure of the paper should have some further annotation of key depositional or morphological features such as ridge crests. Perhaps making the barrier colour grey and using black or red lines to illustrate other features of the barrier morphology would be a way forward. ]]**

+ Referring to the issue with Figure 2, this is partly why Figure 4 is constructed the way it is. Earlier versions of Figure 15 did have ridge crests, but they were very hard to see owing to the narrowness of the barrier.

We add a note to the caption of 15 to compare with ridge crest orientations seen in Figure 4:

"Compare with Figure 4 for detailed morphochronology, including traces of relict ridges."

---

## Author Response (AR2)

**Reviewer #1 Re-Review (Anonymous)**

**[[ I am afraid this revised version of the manuscript with the reference "egusphere-2023-1307" and entitled "Implications for the Resilience of Modern Coastal Systems Derived from Mesoscale Barrier Dynamics at Fire Island, New York" does not show significant improvements with relation to the original version. I can understand to some extent, as the authors received relatively positive feedback during the first round. Yet, some improvements could have been done and were asked to be done to the results section, in relation to the GPR data interpretation, and to the discussion section. ]]**

+ We believe the original comments were constructive and helped improve the manuscript significantly, particularly in terms of framing the Discussion. We are open to (and have made) further modifications based on particulars brought up in this review.

**[[ I continue to find that the work is fine; however, it is not easy to follow as many names to specific locations are continuously referred in the manuscript and the number of figures is high but with different scales of representation and focus areas, so that the reader gets easily lost. This turns makes it difficult to enjoy the reading of the manuscript. As suggested by other reviewers, the authors could have tried to reduce the number of figures. In relation to the figures with the islands, I would recommend trying to condense more the information and if possible use the same limits or scales. Figure 4, for example presents a very different scale than previous figures, why don't you show the same extent as in Figure 2? It does help that the sections in figure 3 are noted in figure 2. ]]**

+ Unfortunately, we cannot use the same limits and scales when relating the full structure of the island (51 km long) versus our individual study areas. The other problem is that our figures are already dense with information, and we believe further condensing would create even more confusion.

      This review also pointed out that aspects of our investigation, such as our GPR profiles, are unnecessary, which would save figure space. However, we strongly disagree with this sentiment (see next comment).

**[[ Regarding the GPR, it is still not clear to me if it is actually needed. The authors made the effort to justify its contribution, however, I still have some doubts as with the interpretations from older maps and morphology, the authors reconstruct the history of the islands with a good level detail. Also, the information that the GPR brings overlaps with the information on evolution you can obtain from the DTM models that you have. ]]**

+ We take this as a compliment that we were able to relate our desktop study of the landscape so well that it seemed we did not even need even to perform a GPR investigation to confirm our suspicions. However, our use of GPR follows standard protocols for investigating the evolution of a barrier island. As Dougherty et al. (2019) states, "once the surface morphology is analysed, the next step to determine how a barrier formed is to study the history preserved in the shallow subsurface. The lidar data should be used to make informed decisions on where best to acquire detailed stratigraphy using geophysics." Without GPR, it would be impossible to understand the full lateral extents and thicknesses of various deposits, which is information we used to help guide

the placement of our sediment cores. Ultimately, the combination of cores and GPR allowed us to chronologically constrain the barrier's geomorphology and provide insight into the types of environments the island previously supported.

Reference:
Dougherty, A.J., Choi, J.H., Turney, C.S., & Dosseto, A: Optimizing the utility of combined GPR, OSL, and Lidar (GOaL) to extract paleoenvironmental records and decipher shoreline evolution, Climate of the Past, 15(1), 389-404, 2019. https://cp.copernicus.org/articles/15/389/2019/

**[[ Besides, and as suggested in my previous comments, I still believe the data processing and interpretation could improve. It is not clear what the authors try to highlight in the interpreted sections, as they do not define clear patterns and radar facies. The use of black lines in the interpreted sections are intended to note the presence of prominent reflections related to washover and beach sediments. This should be better explained as sometimes the lines seem the result of a rather random interpretation of the radargrams. Are the lines intended to separate major units? Like major boundary surfaces related to erosive events? Or do they show the internal configuration of facies? ]]**

+ Highlighted reflections show the internal configuration of the facies in the subsurface. To make this distinction clear, and to highlight the extents of individual facies types, we now add consistent color highlight on the inferred/discernable limits of facies. Legends are provided in figures to aid interpretation.

**[[ I still do not understand very well the relationship between the water table (figure 7) and the mean sea level. It seems that the MSL is close to 0 m (NAVD88), however the WT is 2.5 m below this level, is that normal? It does not seem logic at all. Did you interpret the WT based on ground truth observation? I would revisit this as it seem a bit odd. Also, considering the fact that the barrier is so narrow in some areas, it is even easier to imagine a high water table, also supported by the penetration of salt or marine waters across the barrier, this also shows the intrusion above the level of 0 m. I am asking about this because the elevation of the WT determines the model for depth conversion from nano seconds to meters. Usually, the presence of water reduces the velocity to half of it and that would imply that the depth below that WT level should be expanded and for instance, reflections interpreted as being close to 5 m depth would be around 2.5 m. Therefore, I insist on making sure this is not an issue. If the WT is where it is usually (around MSL) the depth correction should be different above and below this level. ]]**

+ "Water Table" should be "Saltwater Table". There is a freshwater lens sitting on top of that boundary, which created some challenges for the sand auger. We are reasonably confident in our velocities inferred from hyperbola analysis since the linearly decompacted cores show good correlation with the facies placement seen in our GPR profiles. Additionally, our GPR lines over Whalers Inlet located a reflection at around -5 m which is consistent with the depth of the antecedent Pleistocene surface as reported by Schubert (2010) and mentioned in the manuscript.

Please note that the method we employ to for determining an average radar wave velocity from hyperbola analysis has been used by other authors:

Hede, M.U., Bendixen, M., Clemmensen, L.B., Kroon, A., and Nielsen, L.: Joint interpretation of beach-ridge architecture and coastal topography show the validity of sea-level markers observed in ground-penetrating radar data, The Holocene, 23(9), 1238-1246, 2013.

Layek, M.K., Debnath, P., Sengupta, P., & Mukherjee, A.: Delineation of sedimentary facies and groundwater-sea water disposition in an intertidal zone of the Bay of Bengal using GPR and VES, Journal of Environmental and Engineering Geophysics, 23(2), 235-249, 2018.

**[[ Another issue that I would like the authors to explain better is the problem with the age of the peat or organic horizons and the disparity with the plant materials. Also, it would be interesting to elaborate on the increase in age seaward from C1 to C3, also, how do you explain the occurrence of overwash processes in C1 and C3? If I understand this well, if C3 is overwashed, that would mean that C1 could hardly be as they are about 300 m apart. But even more difficult, is to understand the age or moment when that would happen as the organic level in C1 is half the age than in C3. Also, considering the GPR record, this is not easy to imagine. In the GPR line in figure 11, where these cores are located, the authors include dashed black lines to note dune surface buried by beach facies. Could the authors explain a bit better what do they mean with that and the associated implications? I imagine at least a large gap in the sedimentary record if the dunes are developed on top of beach sediments as suggested from the GPR lines to explain the jump in the sea level of about 2 m. Also, make sure the reader can understand those cases where an overwash buries a dune that formed above a beach without eroding it. ]]**

+ The disparity with the age of the peat versus plant remains arises because of root intrusion into existing peat deposits. All the radiocarbon samples analyzed for this study showed evidence of well-preserved roots/rootlets, so plant remains are expected to be generally younger than the bulk sediment age. The only sample where this age disparity was not present was in Core C9, where the organic sediment and plant ages overlapped in time. That particular sample comes from a thin organic horizon that probably represents an exceptional case (for Fire Island), where marsh was very quickly created and destroyed due to inlet proximity (Long Cove Inlet).

Regarding the issue at Point O' Woods, the interpretation of the reviewer is correct. There is a large gap in time due to the area around Point O' Woods being truncated by erosion at some point, probably after progradation well seaward of C3's location. As for the overwash noted at the rear of the island, past storms have been able to spot breach the relict dune lines (which are now topographically subdued due to island drowning). This is evident in the morphology, and Figure 3d highlights some prominent washover channels that penetrate the barrier from front to back. Additionally, post-storm imagery from the 1930s to the present also shows spot breaching in areas where relict dunes are topographically subdued.

**[[ Then, one more major issue with the manuscript is related to the discussion. I had suggested the authors not to insist on the unknowns, mostly because I found it too speculative and reduced on their interpretations of what could happen by not including or considering other possible processes and thus, trajectories. However, the authors decided not to change the document following the suggestion. As a result, this subsection of the discussion is mostly**

**filled with conjectures and with the strange believe that having a large dune is negative for the resilience of a coastal barrier. ]]**

+ We discuss the 'unknowns' because the morphological trends we highlight on Fire Island (narrowing, loss of marshes, drowning of barrier interior) have been seen in modeling and historically observed. These negative effects associated with large dunes are discussed by Timmons et al. (2010) and Dolan (1972). Additionally, the modeling by Mariotti and Hein (2022) demonstrating lag dynamics resulting from geomorphic capital are an extension of this.

Regarding the issue of resilience versus resistance, large dunes specifically increase barrier island *resistance* to landscape change over short-term annual to decadal scales. Counterintuitively, this short-term resistance comes at a price for barrier island *resilience* over longer timescales (decadal to centennial) by limiting (or eliminating) overwash that is required to build height/width while translating the barrier landward so that it can keep up with rising sea levels. In this context, because Fire Island remains mostly static, its backbarrier ecosystems are being degraded by erosion and drowning. In addition to slowly breaking down the island's ecological resilience, erosion and drowning is narrowing the barrier and potentially making it less physically resilient to future transgression in a time of increasing sea level and maybe increasing storm frequency/intensity.

To address the lack of clarity regarding both points, significant modifications were made to the Discussion, which are noted in the response to the reviewer's final comment (see below).

**[[ Well, this is very strange to read and only based on a narrow view of the evolution of a system. The authors assume that dunes can only be eroded, they do not consider the possibility that these features may adapt and even contribute sand to the backbarrier. For the authors, barrier inland migration can only occur driven by overwash. If a dune is able to build up, it is because sediment budget is high (as in this case helped by human interference on the evolution of tidaly inlets) and thus, it is resilient. If a change in the conditions happens, then, the shoreline will react and thus will do the dune. There are examples showing how dunes are able to move landward while maintaining or even gaining elevation and transferring sediment inland (example in some work by Davidson-Arnott and colleagues). Of course, this is not possible along armored coastal stretches, but also not considered at all as a possibility by the authors of this work. Then, all of a sudden we end understanding that only a coastal barrier on a unstable state (usually associated with a rollover phase) is more resilient than a barrier with a tall dune that it is still unknown which adaptation trajectory may follow. ]]**

+ This crux of this comment oversimplifies the dynamics at play, cites examples of transgressional dunes that may not be relevant in this system, and ignores many other well-documented cases where large dune systems have been completely destroyed over decadal timescales. The Virginia Eastern Shore barrier islands, for example, offer multiple cases of dunes and almost entire islands being lost (e.g. Hog Island and Cobb Island, among others). Additionally, this comment conflates the idea of 'resistance' with 'resilience', which we addressed in the response to the previous comment. As we noted, Fire Island's very tall dunes make it resistant to change, but not necessarily resilient in the long term. This loss of resilience is already being realized in sections of the barrier that have remained static for hundreds of years. We modified the Discussion heavily based on the

previous comment, which addresses the concerns here. Our modifications are shown in response to the reviewer's final comment (see below).

[[ In this regard, I must confess that I very much regret my initial positive comments to the manuscript, as I can hardly understand this discussion. In this regard, I suggest considering rearranging it. I do believe the noted change in state is worth studying and paying attention to understand the evolution of the system, but I do not understand the suggestion that the system may be close to a tipping point. ]]

+ Ultimately, the reviewer's insights proved helpful to improving the manuscript, although this most recent round of comments demonstrated our Discussion required further refinement. We believe our revisions to the Discussion (see below, in response to final comment) should clear up the lack of understanding.

[[ Yet, the authors should also consider changing the title of the manuscript, as it is not clear what does it mean. Implications of what? The implications are the unknown, so, do you want to state that in the title? It is like setting a question you cannot answer. The only thing you do know is that the state, and thus the dynamics and processes dominating the morphology (avoid using eco as you do not assess that component), have significantly changed and that may determine the future of the islands, but maybe only that the moment for barrier rollover will have a greater delay. ]]

+ The reviewer correctly points out the main implication of the manuscript, which is that barrier rollover may have a greater delay, and we do not yet understand exactly what that means in the context of long-term resilience for an island which is now experiencing a greater rate of sea-level rise than at any time in the last 5,000 years. We can, however, point out that the short-term trend of bayside and interior drowning is likely to continue due to the presence of large overwash-resistant dunes, and we can also point to modeling studies and observations in other systems to show that a transition to rapid transgression and possibly even island drowning is possible. As our title states, these will have implications for resilience, and over multiple timescales.

[[ Still in the title, what do you mean by coastal systems? Maybe it is too broad. ]]

+ We currently qualify this with "Barrier Dynamics" in the title. An alternative could be to write "Coastal Barrier Systems," but we decided against this due to redundancy.

[[ In the introduction section, lines 33 to 35, I would recommend the authors to revise this definition if they are actually using the one chosen by Masselink and Lazarus. The authors present a large discussion about the difference between the engineering or static approach to resilience and the ecologic/dynamic one, choosing the second to define the resilience of natural and not so natural systems, as the ones able to respond to disturbances maintaining their functions. I would recommend the authors to also include those approaches where dunes are included within the system, adding complexity and processes but more aligned with what their study area presents. ]]

+ We add context to clarify our definition in the first paragraph of the Introduction: "In a general sense, Masselink and Lazarus (2019) define coastal resilience as the capacity of the landscape to maintain its ecological functions in the presence of disturbance. Here, we focus this definition and describe morphologic resilience as the ability of coastal barriers to redistribute sediment in a way that maintains the integrity and distribution of landscape components through time (Masselink and Lazarus, 2019 *c.f.* Long et al., 2006), with state shifts comprising threshold changes in system morphology that cannot be easily recovered to a previous configuration (Kombiadou et al., 2019)."

**[[ In the results section, I found difficult to understand the vertical relationships between the facies mentioned by the authors. So, how can dunes be overwashed? If I imagine the second to happen at the beach level…how are dunes buried by overwash? Then, if sea level is rising across your record, would you expect to be able to note the gradual rise? ]]**

+ It is confusing due to the long timescales involved. Since sea level is increasing, older dunes produced at a lower sea level can be overwashed due to an increase in the surface elevation of the beach. Additionally, washovers channelize over the dune and do not always cut down to the level of the sea in calm conditions. So, together, this allows for some preservation of the dune base beneath washover facies.

**[[ Make sure the length of the cores is well represented within the GPR lines. ]]**

+ Annotations added noting the decompacted core lengths.

**[[ Are the lines in figure 7 running parallel? If so, why do you need to show both? ]]**

+ This was done mostly to show the loss of the foredune in profile from 2016 to 2021, which is now more relevant to the Discussion based on our most recent edit. We use this comparison to demonstrate the concept of resistance.

**[[ In the discussion, the whole paragraph from line 533 to line 549 is very difficult to follow and understand and again, it presents a barrier that is passive and not able to react at all. Why do the authors assume that the current resilience of the island is low? Why it has to continue decreasing? This sentence is impossible to understand: Loss of morphological resilience is likely to be exacerbated by reserves of naturally available sediment in reaches of relict progradational dunes, which could enhance Fire Island's short-term resistance. I would think all the way around, because this means that the width of the barrier is high, how is that a not resilience system? ]]**

+ The width of the barrier is not necessarily high in places where there are multiple dunes present. In fact, it is quite narrow in places like Davis Park, where there at least two to three dune ridges present (see Figure 9; compare with Figure 3 for plan view). However, the barrier (especially the modern foredune) in such places is tall, and due to dunes blocking washover deposition, there is no longer a fringing marsh, and the bay is directly eroding into the base of the most landward relict dunes, destroying terrestrial ecosystems. In this sense, the dunes themselves may be resilient (for now) and provide a level of resistance against rapid landscape change, but at the scale of the whole barrier island the lack of washover is an impediment to the barrier's ability to adapt to changing

environmental conditions. Thus, Fire Island could not be described as a resilient coastal system per the definition of Masselink and Lazarus (2019)—Fire Island's landscape is presently not adapting to long-term disturbance in a way that allows it to maintain its functions.

Since the reviewer's comments make it clear the Discussion was unsuccessful in relating this critical point, we attempt to fix this by breaking up the third paragraph of the Discussion into two separate paragraphs and completely rewriting them. Here, the first paragraph of the reworked section defines why we know Fire Island is presently experiencing a loss of resilience and that we think this process is likely to continue. The second paragraph explains why this loss of resilience is being enhanced by short-term resistance (which is defined with an example in Ho-Hum Beach):

"Regardless of the processes interacting or competing with inlet activity, our results suggest that if Fire Island were capable of sustaining inlets at the decadal scale—as it once did—it would display periodic barrier rotation accompanied by a greater prevalence of overwash. The latter is the primary process sustaining barrier island adaptability to disturbance over decadal timescales (Masselink and Lazarus, 2019). Instead, as sea level increases, the modern island with its nearly continuous high-foredune system is more likely to undergo gradual frontal erosion combined with drowning of the interior and bayside, which is already underway (Art, 1976; Nordstrom and Jackson, 2005; Sirkin, 1972). This process is well-documented historically (Dolan, 1972) and in modeling studies (Lorenzo-Trueba and Mariotti, 2017, Miselis and Lorenzo-Trueba, 2017; Magliocca et al., 2011; Rogers et al, 2015), the latter demonstrating that the lack of washover deposition from high, maintained foredunes can perpetuate a feedback loop of island narrowing and marsh destruction.

Although counterintuitive, loss of morphological resilience at the decadal scale may be exacerbated by reserves of naturally available sediment in places where relict progradational dunes exist, as well as by shallow antecedent topography. In the short term (years to decades), both features likely enhance Fire Island's resistance (see Kombiadou et al. 2019) against landward migration. However, over longer timescale (decades to centuries), modeling by Mariotti and Hein (2022) demonstrates that sediment reserves or "geomorphic capital" may increase the barrier's long-term commitment to retreat, as well as the rate of retreat when it eventually occurs. Shallow antecedent topography can also temporarily pin the barrier in place, further contributing to a long-term lag in sea-level driven retreat (Shawler et al., 2021a). At Fire Island, Ho-Hum Beach appears to provide an example of resistance imparted by both large dunes and shallow antecedent topography. Specifically, Figure 7 shows how the foredune at Ho-Hum Beach in 2016 provided resistance to barrier retreat from shoreline erosion induced by Wilderness Inlet. By 2021, the foredune was destroyed by the inlet's downdrift erosion shadow, but the geomorphic capital provided by this foredune and the relict ridge behind it resulted in frontal erosion of the barrier rather than full-scale migration. Additionally, Figure 7 reveals that the antecedent Pleistocene surface may be relatively shallow at -3 m elevation, which suggests the barrier could be at least partly pinned at this location."